# PEER: A Comprehensive and Multi-Task Benchmark for Protein Sequence Understanding

**Minghao Xu**[1,2] *   **Zuobai Zhang**[1,2] *   **Jiarui Lu**[1,2]   **Zhaocheng Zhu**[1,2]
**Yangtian Zhang**[3]   **Chang Ma**[4]   **Runcheng Liu**[5]   **Jian Tang**[1,6,7] †

*equal contribution   †corresponding author

[1]Mila - Québec AI Institute   [2]Université de Montréal   [3]Shanghai Jiao Tong University
[4]Peking University   [5]Tsinghua University   [6]HEC Montréal   [7]CIFAR AI Research Chair

**contacts:** <minghao.xu, zuobai.zhang>@mila.quebec,  jian.tang@hec.ca

## Abstract

We are now witnessing significant progress of deep learning methods in a variety of tasks (or datasets) of proteins. However, there is a lack of a standard benchmark to evaluate the performance of different methods, which hinders the progress of deep learning in this field. In this paper, we propose such a benchmark called **PEER**, a comprehensive and multi-task benchmark for **P**rotein s**E**quence und**ER**standing. PEER provides a set of diverse protein understanding tasks including protein function prediction, protein localization prediction, protein structure prediction, protein-protein interaction prediction, and protein-ligand interaction prediction. We evaluate different types of sequence-based methods for each task including traditional feature engineering approaches, different sequence encoding methods as well as large-scale pre-trained protein language models. In addition, we also investigate the performance of these methods under the multi-task learning setting. Experimental results show that large-scale pre-trained protein language models achieve the best performance for most individual tasks, and jointly training multiple tasks further boosts the performance. The datasets and source codes of this benchmark are all available at `https://github.com/DeepGraphLearning/PEER_Benchmark`.

## 1 Introduction

Proteins are working horses of life and play a critical role in many biological processes. Understanding the functions of proteins is therefore important in a variety of applications [105, 53, 8]. Thanks to the recent progress of protein sequencing [52, 51], a large number of protein sequences are available. For example, in the UniProt database [13], more than 200 million protein sequences are available. This offers a big opportunity for machine learning methods to analyze these sequences and understand the functions of proteins.

Indeed, a variety of deep learning methods have been successfully applied to different protein tasks such as protein function prediction [24, 93, 104], protein structure prediction [36, 4] and protein engineering [53, 8]. These works generally borrow techniques from the natural language processing (NLP) community to learn effective protein representations with different sequence encoders, *e.g.*, CNNs [74], LSTMs [63] and Transformers [63, 19, 65]. However, these approaches are usually evaluated on different tasks/datasets, and there lacks a standard benchmark to systematically evaluate the performance of different techniques, which hinders the progress of deep learning for protein understanding.

In this paper, we are inspired by the success of deep learning in computer vision and natural language processing community, where, to a large extent, the progresses are driven by benchmark datasets such as ImageNet [17] and GLUE [90]. Therefore, we take the initiative of building a comprehensive

36th Conference on Neural Information Processing Systems (NeurIPS 2022) Track on Datasets and Benchmarks.

benchmark for **P**rotein s**E**quence und**ER**standing (**PEER**), to facilitate the progress of deep learning in protein understanding. The PEER benchmark includes seventeen biologically relevant tasks that cover diverse aspects of protein understanding, including protein function prediction, protein structure prediction, protein localization prediction, protein-protein interaction prediction and protein-ligand interaction prediction. In each benchmark task, the training/validation/test splits are carefully designed to evaluate the generalization ability of deep learning techniques in real-world settings. For example, in protein engineering tasks, low-order mutants of wild type sequences are used as training data while high-order mutants are used as testing data to evaluate the out-of-distribution generalization ability of deep learning approaches.

For each individual task, we evaluate the performance of different types of sequence-based approaches, including traditional feature engineering approaches, different protein sequence encoders such as CNNs, LSTMs and Transformers, and large-scale pre-trained protein language models. In addition, we also expect to develop general approaches that can perform well across different protein tasks and benefit from knowledge sharing and transferring. We therefore also evaluate different approaches under the multi-task learning setting. Experimental results show that pre-trained protein language models achieve the best performance on most individual tasks by either directly using the pre-trained protein language models as sequence encoders or fine-tuning them with task-specific supervised data. Jointly training multiple tasks can further enhance the performance, showing the potential of multi-task learning.

We maintain the leaderboards of PEER benchmark at `https://torchprotein.ai/benchmark`, where we will receive new benchmark results from the community in the near future. We hope this benchmark will serve as a good starting point and spark the interest of machine learning community in working on deep learning for protein understanding. In the future, we will further extend the current benchmark by going beyond sequence-based approaches to structure-based approaches.

## 2   Related Work

**Protein representation learning.** Protein representation learning has been studied for decades and recently drawn increasing interest from different domains. Early methods on protein representation learning define different rules to extract physiochemical or statistical features from protein sequences [42, 43, 70, 20, 92]. Advancements of deep learning and natural language processing motivate the development of protein sequence models to utilize large-scale protein sequence corpus. Early works along this line adapt the idea of word2vec [55] and doc2vec [46] to protein sequences [89, 40, 97, 54]. To increase model capacity, deeper sequence encoders originally developed by the NLP community are pre-trained on million- or billion-scale protein sequences. Well-known works include UniRep [1] and ProtXLNet [19], which are pre-trained with the next amino acid prediction task, and TAPE Transformer [63], ProtBert [19], ProtAlbert [19], ProtElectra [19], ProtT5 [19] and ESM [65] with masked language modeling task, PMLM with pairwise masked language modeling loss [28] and CPCProt with contrastive predictive coding loss [50]. Recently, there are also some works exploring protein multiple sequence alignments (MSAs) [64, 8, 53], 3D structures [30, 104] and surfaces [22, 85, 15] for learning effective protein representations. The structure-based approaches outperform sequence-based approaches on some tasks, but sequence-based approaches still dominate the performance for most tasks thanks to the large number of protein sequences. Therefore, in this benchmark, we mainly focus on sequence-based approaches.

**Protein modeling benchmarks.** To fairly compare different protein modeling methods, computational biologists have devoted a lot of effort to build large-scale and comprehensive benchmarks. Among them, the best-known one is the biennial Critical Assessment of protein Structure Prediction (CASP) [44], which focuses on protein structure prediction and becomes a golden-standard assessment in this area. Accompanied with CASP, the Critical Assessment of Functional Annotation (CAFA) challenge [105] is held for the evaluation of protein function prediction. To comprehensively compare different machine learning methods, the TAPE benchmark [63] is built on five tasks spread across different domains of protein biology and evaluate the performance of protein sequence encoders. FLIP [16] proposes three protein landscape benchmarks for fitness prediction evaluation. A recent work [88] focuses on the evaluation of unsupervised protein representations and evaluates 23 typical methods. TDA [34] contains protein-related datasets and tasks for drug discovery. ATOM3D [86] provides benchmark datasets for 3D structure based biomolecule understanding. Another recent work [10] builds a benchmark containing 7 downstream tasks for self-supervised protein representation learning.

Table 1: Benchmark task descriptions. Each task, along with its acronym, category, the source of dataset, the number of unique proteins, *mean (std)* of protein sequence lengths, the size of each split and evaluation metric are shown below. *Abbr.*, Reg.: regression; Cls.: classification; Acc: accuracy; RMSE: root-mean-square error; Seq. len.: sequence length.

| Task (Acronym) | Task Category | Data Source | #Protein | Seq. len. | #Train/Validation/Test | Metric |
|---|---|---|---|---|---|---|
| **Function Prediction** | | | | | | |
| **GB1 fitness prediction (GB1)** | Protein-wise Reg. | FLIP [16] | 8,733 | $378.6_{(0.9)}$ | 381/43/8,309 | Spearman's $\rho$ |
| **AAV fitness prediction (AAV)** | Protein-wise Reg. | FLIP [16] | 82,583 | $1033.0_{(3.4)}$ | 28,626/3,181/50,776 | Spearman's $\rho$ |
| **Thermostability prediction (Thermo)** | Protein-wise Reg. | FLIP [16] | 7,158 | $880.6_{(974.2)}$ | 5,149/643/1,366 | Spearman's $\rho$ |
| **Fluorescence prediction (Flu)** | Protein-wise Reg. | Sarkisyan's dataset [71] | 54,025 | $343.3_{(1.3)}$ | 21,446/5,362/27,217 | Spearman's $\rho$ |
| **Stability prediction (Sta)** | Protein-wise Reg. | Rocklin's dataset [66] | 68,934 | $66.6_{(5.2)}$ | 53,571/2,512/12,851 | Spearman's $\rho$ |
| **$\beta$-lactamase activity prediction ($\beta$-lac)** | Protein-wise Reg. | Envision [25] | 5,198 | $396.1_{(0.7)}$ | 4,158/520/520 | Spearman's $\rho$ |
| **Solubility prediction (Sol)** | Protein-wise Cls. | DeepSol [39] | 71,419 | $424.1_{(225.9)}$ | 62,478/6,942/1,999 | Acc |
| **Localization Prediction** | | | | | | |
| **Subcellular localization prediction (Sub)** | Protein-wise Cls. | DeepLoc [2] | 13,961 | $665.3_{(395.3)}$ | 8,945/2,248/2,768 | Acc |
| **Binary localization prediction (Bin)** | Protein-wise Cls. | DeepLoc [2] | 8,634 | $636.5_{(396.5)}$ | 5,161/1,727/1,746 | Acc |
| **Structure Prediction** | | | | | | |
| **Contact prediction (Cont)** | Residue-pair Cls. | ProteinNet [3] | 25,563 | $320.0_{(275.2)}$ | 25,299/224/40 | L/5 precision |
| **Fold classification (Fold)** | Protein-wise Cls. | DeepSF [31] | 13,766 | $235.4_{(155.1)}$ | 12,312/736/718 | Acc |
| **Secondary structure prediction (SSP)** | Residue-wise Cls. | NetSurfP-2.0 [41] | 11,361 | $360.5_{(229.3)}$ | 8,678/2,170/513 | Acc |
| **Protein-Protein Interaction Prediction** | | | | | | |
| **Yeast PPI prediction (Yst)** | Protein-pair Cls. | Guo's dataset [26] | 1,707 | $726.3_{(432.0)}$ | 1,668/131/373 | Acc |
| **Human PPI prediction (Hum)** | Protein-pair Cls. | Pan's dataset [59] | 5,553 | $727.7_{(438.2)}$ | 6,844/277/227 | Acc |
| **PPI affinity prediction (Aff)** | Protein-pair Reg. | SKEMPI [56] | 627 | $304.9_{(193.8)}$ | 2,127/212/343 | RMSE |
| **Protein-Ligand Interaction Prediction** | | | | | | |
| **Affinity prediction on PDBbind (PDB)** | Protein-ligand Reg. | PDBbind [49] | 10,607 | $414.9_{(234.3)}$ | 16,436/937/285 | RMSE |
| **Affinity prediction on BindingDB (BDB)** | Protein-ligand Reg. | BindingDB [47] | 1,006 | $799.8_{(417.0)}$ | 7,900/878/5,230 | RMSE |

These previous benchmarks mainly focus on specific types of protein modeling tasks, which are insufficient to evaluate the versatility and general effectiveness of a protein encoder. In this work, we propose the comprehensive PEER benchmark, including tasks for protein function/localization/structure prediction, protein-protein interaction prediction and protein-ligand interaction prediction. Besides benchmarking under conventional single-task learning, we additionally evaluate the impact of multi-task learning on protein sequence modeling, which has shown great potential in the NLP community according to the GLUE [90] and SuperGLUE [91] benchmarks.

## 3 Benchmark Tasks

The PEER benchmark includes 17 benchmark tasks within 5 task groups in total. We represent a protein $x$ as a sequence of amino acids (*a.k.a.*, residues) $x = (x_1, x_2, \cdots, x_L)$ of length $L$. For protein-ligand interaction prediction, we represent a ligand $g$ as a molecular graph $g = (\mathcal{V}, \mathcal{E})$, where $\mathcal{V}$ and $\mathcal{E}$ denote the atom set and the bond set, respectively. For each task, we list the task name and its acronym, task category, data source, protein sequence statistics, dataset statistics and evaluation metric in Tab. 1.

### 3.1 Protein Function Prediction

This group of tasks intend to predict functional values of proteins (either discrete or continuous). We select tasks from important protein engineering applications, and the data splits are designed to evaluate the performance of machine learning methods in real-world scenarios.

**GB1 fitness prediction** measures the fitness values of possible mutants of the GB1 protein, with the target $y \in \mathbb{R}$ collected by Wu et al. [95]. We adopt the "2-vs-rest" dataset splits proposed in FLIP [16], where the training and validation sets consist of wild type, single and double mutants, while others are assigned to the test set. This landscape focuses on combinations of mutations at four epistatic sites and thus can test the ability of the model to model epistatic interactions.

   *Impact*: The protein G, an immunoglobulin binding protein, resorts to the GB1 binding domain to act its function [72, 77]. This task investigates the effects of interactions between mutations and how to improve the fitness of this important functional protein via engineering.

**AAV fitness prediction** tries to evaluate the fitness scores $y \in \mathbb{R}$ of mutants of VP-1 AAV proteins. The rich mutational screening landscape is collected by mutagenizing a 28-amino acid window from position 561 to 588 of VP-1 [9]. We adopt the "2-vs-rest" dataset splits from FLIP [16], which includes mutants with two or less mutations in the training and validation sets and others in the test

set. The models are expected to effectively capture the mutational effects in a specific region of a large protein.

*Impact*: The engineering of Adeno-associated virus (AAV) capsid proteins is of great gene therapy interest, since it can assist the virus in integrating a DNA payload into a target cell. This task studies how to predict the fitness for a long sequence being mutated in a specific region and thus help engineering applications of large proteins.

**Thermostability prediction** investigates the thermostability 48,000 proteins across 13 species. The target $y \in \mathbb{R}$ corresponds to protein melting curves measured by a mass spectrometry-based assay, which are curated from the Meltome Atlas [35]. The "human-cell" split protocol in FLIP [16] is adopted in our benchmark, where the authors cluster sequences of one cell line for human and assign cluster representatives to training and test sets. This task tests the capacity of the model on landscapes including both global and local variations.

*Impact*: Thermostability is a desirable feature complementing with other application-specific functions, which enables operations at higher reaction temperatures with faster reaction rates [45]. This task can boost protein engineering applications by developing a good thermostability predictor.

**Fluorescence prediction** asks the model to predict the fitness of green fluorescent protein mutants. The target $y \in \mathbb{R}$ is the logarithm of fluorescence intensity annotated by Sarkisyan et al. [71]. We adopt the dataset splits from TAPE [63], where the training and validation sets consist of mutants with three or less mutations, and the test set is composed of mutants with four or more mutations. This task measures the transferability of the model from training on lower-order mutants to evaluating on higher-order mutants.

*Impact*: Green fluorescent protein is an important marker protein, enabling scientists to see the presence of the particular protein in an organic structure by its green fluorescence [87]. This task could reveal the mutational patterns of enhancing/reducing such biological property.

**Stability prediction** attempts to evaluate the stability of proteins under natural environment. The target $y \in \mathbb{R}$ indicates the experimental measurement of stability. We adopt the dataset from Rocklin *et al.* [66] and follow the dataset splits of TAPE [63], where the proteins from four rounds of experimental design are used for training and validation, and the top candidates with single mutations are used for test. This task evaluates the generalization ability by training on the data with multiple mutations and applying to discover the top candidates with single mutations.

*Impact*: The stability of a protein affects whether its function can be carried out in the body [75]. This benchmark task simulates the real-world application scenario of selecting functional mutants that possess decent stability.

$\beta$-**lactamase activity prediction** studies the activity among first-order mutants of the TEM-1 beta-lactamase protein. The target $y \in \mathbb{R}$ is the experimentally tested fitness score which records the scaled mutation effect for each mutant. The sequences of mutants along with their labels are taken from Envision [25], and the dataset split protocol follows Rives et al. [65]. The models with high capacity are expected to discriminate proteins with only one-residue difference in the dataset.

*Impact*: TEM-1 beta-lactamase is the most widespread enzyme that endows gram-negative bacteria with beta-lactam antibiotic resistance [58]. This task studies how to enhance the activity of this important enzyme by single mutations.

**Solubility prediction** aims to predict whether a protein is soluble or not (*i.e.*, with label $y \in \{0, 1\}$). We adopt the training, validation, and test splits from DeepSol [39], where protein sequences with sequence identity $\geqslant 30\%$ to any sequence in the test set are removed from the training set. This task evaluates a model's ability to generalize across dissimilar protein sequences.

*Impact*: Protein solubility plays a critical role in pharmaceutical research and industry field, since good solubility is an essential property for a functional protein [39]. This task aims to boost the development of effective in silico sequence-based protein solubility predictor.

## 3.2 Protein Localization Prediction

The localization of proteins is highly related to their *in vivo* functionality. This task group contains two levels of protein localization prediction.

**Subcellular localization prediction** expects the model to predict where a natural protein locates in the cell. For example, the proteins naturally existing in the lysosome will be attached with a categorical

label *"lysosome"*. There are 10 possible localizations, inducing the label $y \in \{0, 1, \cdots, 9\}$. We adopt the training and test sets introduced in DeepLoc [2]. In DeepLoc, homologous protein sequences are clustered with 30% sequence identity and split into five folds. Four of them are used for training, and the rest one is held out for testing. We randomly split out a validation set from the training set with a 4:1 training/validation ratio. This task evaluates the ability of the model to correctly predict the subcellular localization of homologous proteins.

*Impact*: Acquiring the subcellular localization of a protein can greatly improve target identification during drug discovery [62]. A high-throughput and accurate prediction tool of subcellular localization can accelerate this whole process. This task facilitates the development of such a tool.

**Binary localization prediction** is a much simpler version of the task above, where a model is asked to coarsely classify each protein to be either "membrane-bound" or "soluble" (*i.e.*, with label $y \in \{0, 1\}$). The training and test sets are also from DeepLoc [2], where we retain the samples attached with binary localization labels. We randomly hold out a validation set from training with a 4:1 training/validation ratio. This task also evaluates the generalization across homologous proteins.

*Impact*: The "soluble" proteins are free molecules in the body, while the "membrane-bound" proteins may contain some catalytic activity by binding to the membrane [23]. This task boosts the efficient discrimination of these two types of proteins by machine learning techniques.

### 3.3 Protein Structure Prediction

The accurate prediction of protein folding structures is critical to understand their various functions. This group involves three sub-problems of general protein structure prediction.

**Contact prediction** estimates the contact probability of each pair of residues, where each residue pair is associated with a binary label $y \in \{0, 1\}$ indicating whether they contact (*i.e.*, within a distance threshold $\delta$) or not. We adopt the ProteinNet dataset [3] for this task. Following TAPE [63], we use the ProteinNet CASP12 test set for evaluation, which is filtered against the training set at a 30% sequence identity. According to the standard of CASP [57], we report the precision of the L/5 most likely contacts for medium- and long-range contacts on the test set. Such evaluation measures the ability of a contact prediction model on predicting the folded structures of diverse protein sequences.

*Impact*: The prediction of amino acid contacts from protein sequence is a crucial step towards the prediction of folded protein structures [7]. The evaluation of this task pays particular attention to medium- and long-range contacts for their critical roles in protein folding.

**Fold classification** classifies the global structural topology of a protein on the fold level, represented as a categorical label $y \in \{0, 1, \cdots, 1194\}$. The label is determined by the backbone coordinates of the corresponding protein structure. We adopt training, validation, and test sets from Hou's dataset [31], originally derived from the SCOP 1.75 database [21]. All proteins of a given fold class are further categorized into related *superfamilies*. Entire superfamilies are held out from training to compose the test set, allowing us to evaluate the ability of the model to detect the proteins with similar structures but dissimilar sequences, *i.e.*, performing remote homology detection [63].

*Impact*: Protein fold classification is important for both functional analysis and drug design [11]. The SCOPe database [21] only categorizes a small portion of proteins in PDB [6]. This task aims to empower automatic fold classification from protein sequences by machine learning.

**Secondary structure prediction** predicts the local structures of protein residues in their natural state. A secondary structure label $y \in \{0, 1, 2\}$ (*i.e.*, coil, strand or helix) is assigned to each residue. We adopt the training set from Klausen's dataset [41], which is filtered such that no two proteins have greater than 25% sequence identity. For test, we use the CB513 dataset [14], and it is filtered at 25% sequence identity against training to evaluate the generalization across dissimilar protein sequences.

*Impact*: The accurate prediction of protein secondary structure is useful in multiple aspects, *e.g.*, protein function understanding [41] and multiple sequence alignment [76]. This benchmark task approaches such a goal by enabling the training and generalization test of machine learning models.

### 3.4 Protein-Protein Interaction Prediction

Protein-protein interaction (PPI) prediction is crucial for protein complex structure modeling and protein function understanding. This group involves three PPI prediction tasks.

**Yeast PPI prediction** predicts whether two yeast proteins interact or not (*i.e.*, with a binary label $y \in \{0, 1\}$). We use Guo's yeast PPI dataset [26], where negative pairs are from different subcellular

locations. For dataset split, we first remove redundancy within all protein sequences in the dataset with a 90% sequence identity cut-off, and then randomly split these filtered sequences into training/validation/test splits. After that, we remove the redundancy between each split pair with a 40% sequence identity cut-off. The generalization across dissimilar protein sequences are thus evaluated.

*Impact*: It is of broad scientific interest to construct complete and precise yeast interactome network maps [101, 61, 5]. The benchmark task will aid to this project by predicting binary yeast protein interactions with machine learning models.

**Human PPI prediction** predicts whether two human proteins interact or not (*i.e.*, with a binary label $y \in \{0, 1\}$). We adopt Pan's human PPI dataset [59] that contains positive protein pairs from Human Protein Reference Database (HPRD) [60] and negative pairs from different subcellular locations. The dataset splitting scheme follows that of yeast PPI prediction, except that a 8:1:1 train/validation/test ratio is adopted here. This task also evaluates generalization across dissimilar protein sequences.

*Impact*: Unraveling the human protein interactome is vital to understand mechanisms of disease and uncover unknown disease genes, motivating many projects [68, 102, 67]. This benchmark task is expected to contribute by boosting effective machine learning models for human PPI prediction.

**PPI affinity prediction** estimates the binding affinity $y \in \mathbb{R}$ measured by $pK_d$ between two proteins. We utilize the SKEMPI dataset [56] and split it according to the number of mutations. In specific, the training set consists of wild-type complexes as well as the mutants with at most 2 mutations; the validation set consists of the mutants with 3 or 4 mutations; the test set is composed of the mutants with more than 4 mutations. Therefore, this task evaluates model's generalization ability under a multi-round protein binder design scenario.

*Impact*: Predicting the relative binding strength among candidate binders is important for protein binder design [48, 73]. This task provides a test field for machine learning models in such a real-world application.

## 3.5 Protein-Ligand Interaction Prediction

Protein-ligand interaction (PLI) prediction seeks to model the interaction strength between pairs of *protein* and *ligand*. We involve two tasks from different data sources here. Both tasks aim to estimate the binding affinity $y \in \mathbb{R}$ measured by $pK_d$.

**PLI prediction on PDBbind** adopts the PDBbind-2019 dataset [49]. We choose the test set according to the CASF-2016 benchmark [81] to evaluate model generalization. To avoid redundancy, we first remove training sequences against test ones with a 90% sequence identity cut-off, and then cluster the training sequences and randomly split the clusters into training/validation splits with a 9:1 ratio. Note that, we use only the refined-set in PDBbind for better binding affinity data quality.

*Impact*: The recognition of the interactions between small molecules and target proteins is a prominent research topic in the field of drug discovery [98, 94]. This benchmark task seeks to assess the ability of machine learning models to accomplish such a goal.

**PLI prediction on BindingDB** adopts the BindingDB dataset [47]. We follow the dataset splitting scheme in DeepAffinity [37], where 4 protein classes (ER, GPCR, ion channels and receptor tyrosine kinases) are held out from training and validation for generalization test.

*Impact*: Similar as the task above, this task is attractive to the drug discovery community, and it focuses on the evaluation of ligands' interactions with four specific classes of proteins.

# 4 Methods

## 4.1 Baselines

We consider three types of baseline models in our benchmark, *i.e.,* feature engineers, protein sequence encoders and pre-trained protein language models. We summarize each model along with its model type, input layer, hidden layers, output layer and number of parameters in Tab. 2.

**Feature engineers.** We adopt two typical protein sequence feature descriptors, *i.e.*, Dipeptide Deviation from Expected Mean (DDE) [70] and Moran correlation (Moran) [20]. The DDE feature descriptor (400 dimensions) is based on the dipeptide frequency within the protein sequence, and it is centralized and normalized by the theoretical mean and variance computed by amino acid codons.

Table 2: Baseline model descriptions. *Abbr.*, Params.: parameters; feats.: features; dim.: dimension; attn.: attention; conv.: convolutional. "-" indicates a nonexistent component for a model.

| Model | Model Type | Input Layer | Hidden Layers | Output Layer | #Params. |
|---|---|---|---|---|---|
| **Feature Engineer** | | | | | |
| **DDE [70]** | MLP | 400-dim. statistical feats. | linear (hidden dim.:512) + ReLU | - | 205.3K |
| **Moran [20]** | MLP | 240-dim. physicochemical feats. | linear (hidden dim.:512) + ReLU | - | 123.4K |
| **Protein Sequence Encoder** | | | | | |
| **LSTM [63]** | LSTM | 640-dim. token embedding (21 entries) | 3 × bidirectional LSTM layers (hidden dim.: 640) | weighted sum over all residues + linear (output dim.: 640) + Tanh | 26.7M |
| **Transformer [63]** | Transformer | 512-dim. embedding (24 entries) | 4 × Transformer blocks (hidden dim.: 512; #attn. heads: 8; activation: GELU) | linear (output dim.: 512) + Tanh upon [CLS] token | 21.3M |
| **CNN [74]** | CNN | 21-dim. one-hot residue type | 2 × 1D conv. layers (hidden dim.: 1024; kernel size: 5; stride: 1; padding: 2) | max pooling over all residues | 5.4M |
| **ResNet [63]** | CNN | 512-dim. token embedding (21 entries) + 512-dim. positional embedding | 8 × residual blocks (hidden dim.: 512; kernel size: 3; stride: 1; padding: 1) | attentive weighted sum over all residues | 11.0M |
| **Pre-trained Protein Language Model** | | | | | |
| **ProtBert [19]** | Transformer | 1024-dim. token embedding (30 entries) + 1024-dim. positional embedding | 30 × Transformer blocks (hidden dim.: 1024; #attn. heads: 16; activation: GELU) | linear (output dim.: 1024) + Tanh upon [CLS] token | 419.9M |
| **ESM-1b [65]** | Transformer | 1280-dim. token embedding (33 entries) | 33 × Transformer blocks (hidden dim.: 1280; #attn. heads: 20; activation: GELU) | mean pooling over all residues | 652.4M |

The Moran feature descriptor (240 dimensions) defines the distribution of amino acid properties along a protein sequence. Following iFeature [12], we retrieve 8 physicochemical properties $\{P^k\}_{k=1}^8$ from AAindex Database [38] to construct the Moran feature descriptor. Upon these two kinds of feature vectors, we use a nonlinear MLP to project them to the latent space for task prediction. More details about these two feature engineers are provided in the supplementary material.

**Protein sequence encoders.** We employ four broadly-studied protein sequence encoders, *i.e.*, LSTM [63], Transformer [63], shallow CNN [74] and ResNet [63]. These encoders aim to capture the short-range interactions (shallow CNN and ResNet) and long-range interactions (LSTM and Transformer) within the protein sequence. The final output layer aggregates the representations of different residues into a protein-level representation for task prediction.

**Pre-trained protein language models.** We also evaluates two representative protein language models pre-trained on large-scale protein sequence corpuses, *i.e.*, ProtBert [19] and ESM-1b [65]. Both models are huge Transformer encoders exceeding the size of BERT-Large [18]. ProtBert is pre-trained on 2.1 billion protein sequences from the BFD database [80] with the masked language modeling (MLM) objective, and ESM-1b is pre-trained on 24 million protein sequences from UniRef50 [84] by MLM. We study two evaluation settings in our benchmark by either (1) learning the prediction head with protein language model parameters frozen or (2) fine-tuning the protein language model along with the prediction head.

## 4.2 Model Pipeline

In general, we utilize three model pipelines to solve different types of tasks in the PEER benchmark.

**Protein function, localization and structure prediction tasks** learn a function $y = f_\theta(x)$ that maps protein $x$ to the label $y$, where $f_\theta$ is parameterized by a sequence-based encoder and an MLP predictor. The predictor is applied upon residue-level embeddings for secondary structure prediction, upon the embeddings of residue pairs for contact prediction and upon protein-level embeddings for all other tasks in these three groups.

**Protein-protein interaction prediction tasks** learn a function $y = f_\theta(x, x')$ that maps a pair of proteins $(x, x')$ to the label $y$, where $f_\theta$ is parameterized by a pair of siamese sequence-based encoders and an MLP predictor defined upon the concatenation of the embeddings of two proteins.

**Protein-ligand interaction prediction tasks** learn a function $y = f_\theta(x, g)$ that maps a protein-ligand pair $(x, g)$ to the label $y$, where $f_\theta$ is parameterized by a protein sequence encoder, a ligand graph encoder and an MLP predictor defined upon the concatenated protein-ligand embedding.

## 4.3 Single-Task Learning *vs* Multi-Task Learning

**Single-Task Learning.** A simple and common strategy to solve these protein-related tasks is to train a model exclusively on one task of interest at a time, known as single-task learning. Given a task $t \in \mathcal{T}$ from the pool $\mathcal{T}$ of benchmark tasks, a task-specific loss function $\mathcal{L}_t$ is defined to measure the

correctness of model predictions on training samples against ground truth labels. The objective of learning this single task is to optimize model parameters to minimize the loss $\mathcal{L}_t$ on this task.

**Multi-Task Learning.** We further study multi-task learning as a way to enhance the generalization ability of the model. In our benchmark, the sizes of the training sets on different tasks could be significantly different. To have a fair comparison with single-task learning under comparable training budget, we focus on the multi-task learning setting with *a center task* and *an auxiliary task*. Specifically, given a center task $t_c$ with loss $\mathcal{L}_{t_c}$ and an auxiliary tasks $t_a$ with loss $\mathcal{L}_{t_a}$, we seek to boost the performance on the center task by leveraging the knowledge learned from the auxiliary task. To achieve this goal, we build our model $f_\theta$ under the regime of hard parameter sharing [69], in which we employ a single protein sequence encoder shared across all tasks, and a ligand graph encoder is additionally involved in protein-ligand interaction prediction tasks. During training, the model parameters are optimized by the joint loss of center and auxiliary tasks: $\mathcal{L} = \mathcal{L}_{t_c} + \alpha\mathcal{L}_{t_a}$, where $\alpha$ is the tradeoff parameter balancing two objectives. The number of training iterations is identical to that of single-task learning on center task, and we only test on center task to ensure fair comparison.

# 5 Experiments

## 5.1 Experimental Setups

**Model setups.** For protein function, localization and structure prediction tasks, given the protein embedding, we apply a 2-layer MLP with a ReLU nonlinearity in between to perform prediction. For protein-protein interaction prediction tasks, upon the concatenation of the embeddings of two proteins, a 2-layer MLP activated by ReLU serves as the predictor. For protein-ligand interaction prediction tasks, we involve an additional Graph Isomorphism Network (GIN) [96] with 4 layers and 256 hidden dimensions as the ligand graph encoder, which follows previous practices [33, 82]. Based on the concatenation of protein and ligand embedding, a 2-layer MLP activated by ReLU is applied for prediction.

**Training setups.** We train all models with three seeds (0, 1 and 2) on each task and report the mean and standard deviation of three runs' results. For each run, we train with an Adam optimizer for 50 epochs on contact and human PPI prediction and for 100 epochs on other tasks. We perform 10 times of validation uniformly along the training process, and the test performance of the best validation epoch model is reported. For each model, we search for its learning rate among $[1 \times 10^{-5}, 2 \times 10^{-5}, 5 \times 10^{-5}, 1 \times 10^{-4}, 2 \times 10^{-4}]$ and its batch size among $[1, 2, 4, 8, 16, 32, 64, 128, 256]$ based on the validation performance on the $\beta$-lactamase activity prediction task, and the searched parameters are applied to all tasks for that model. For fine-tuning ProtBert and ESM-1b, we set their learning rate as one-tenth of that of the MLP predictor. Unless specified, the tradeoff parameter $\alpha$ for multi-task learning is set as 1.0. The fluorescence, stability, $\beta$-lactamase activity, PPI affinity, PDBbind and BindingDB prediction tasks are trained with mean squared error; the solubility, subcellular localization, binary localization, fold, secondary structure, yeast PPI and human PPI prediction tasks are trained with cross entropy loss; the contact prediction task is trained with binary cross entropy loss. For all models except for ESM-1b, we use the full protein sequence as the input on all tasks except AAV and Thermostability, and, because of the intrinsic limit on input sequence length, ESM-1b truncates those sequences with more than 1022 residues by keeping the first 1022 residues. For AAV fitness prediction, we keep the mutation region of the long sequence for all models. For Thermostability prediction, limited by the memory of GPUs, we set the truncation length as 1022 for ESM-1b and ProtBert. All experiments are conducted on 4 Tesla V100 GPUs (32GB). We provide ablation studies for important hyperparameters in the supplement.

## 5.2 Benchmark Results on Single-Task Learning

In Tab. 3, we report the benchmark results for single-task learning, in which we evaluate ten models in three model groups. Based on these results, we have following findings:

- **Statistical features are informative for protein sequence understanding.** On all available tasks, the 2-gram based DDE featurization significantly outperforms Moran, the physicochemical featurization scheme. Such results demonstrate that the statistical characteristics of protein sequence segments can effectively reveal some biological properties of proteins, which is consistent with

Table 3: Benchmark results on single-task learning. We report *mean (std)* for each experiment. We use four color scales of green to denote the first, second, third and fourth best performance among all models; SOTA model performance from literature is in gray; "-" indicates a non-applicable setting.

| Task | Feature Engineer | | Protein Sequence Encoder | | | | Pre-trained Protein Language Model | | | | Literature SOTA |
|---|---|---|---|---|---|---|---|---|---|---|---|
| | DDE | Moran | LSTM | Transformer | CNN | ResNet | ProtBert | ProtBert* | ESM-1b | ESM-1b* | |
| **Function Prediction** | | | | | | | | | | | |
| GB1 | $0.445_{(0.023)}$ | $0.069_{(0.003)}$ | $-0.002_{(0.003)}$ | $0.271_{(0.020)}$ | $0.502_{(0.007)}$ | $0.133_{(0.095)}$ | $0.634_{(0.047)}$ | $0.123_{(0.012)}$ | $0.704_{(0.018)}$ | $0.337_{(0.013)}$ | 0.73 (CARP-640M [99]) |
| AAV | $0.649_{(0.012)}$ | $0.437_{(0.008)}$ | $0.125_{(0.025)}$ | $0.681_{(0.013)}$ | $0.746_{(0.003)}$ | $0.739_{(0.013)}$ | $0.794_{(0.014)}$ | $0.209_{(0.001)}$ | $0.821_{(0.010)}$ | $0.454_{(0.008)}$ | 0.81 (CARP-640M [99]) |
| Thermo | $0.349_{(0.007)}$ | $0.331_{(0.003)}$ | $0.564_{(0.007)}$ | $0.545_{(0.031)}$ | $0.494_{(0.021)}$ | $0.528_{(0.009)}$ | $0.660_{(0.009)}$ | $0.562_{(0.001)}$ | $0.669_{(0.028)}$ | $0.674_{(0.002)}$ | 0.78 (ESM-1v [16]) |
| Flu | $0.638_{(0.003)}$ | $0.400_{(0.001)}$ | $0.494_{(0.071)}$ | $0.643_{(0.005)}$ | $0.682_{(0.002)}$ | $0.636_{(0.021)}$ | $0.679_{(0.001)}$ | $0.339_{(0.003)}$ | $0.679_{(0.002)}$ | $0.430_{(0.002)}$ | 0.69 (Shallow CNN [74]) |
| Sta | $0.652_{(0.033)}$ | $0.322_{(0.011)}$ | $0.533_{(0.101)}$ | $0.649_{(0.056)}$ | $0.637_{(0.010)}$ | $0.126_{(0.094)}$ | $0.771_{(0.020)}$ | $0.697_{(0.013)}$ | $0.694_{(0.073)}$ | $0.750_{(0.010)}$ | 0.79 (Evoformer [32]) |
| $\beta$-lac | $0.623_{(0.019)}$ | $0.375_{(0.008)}$ | $0.139_{(0.051)}$ | $0.261_{(0.015)}$ | $0.781_{(0.011)}$ | $0.152_{(0.029)}$ | $0.731_{(0.226)}$ | $0.616_{(0.002)}$ | $0.839_{(0.053)}$ | $0.528_{(0.009)}$ | 0.89 (ESM-1b [74]) |
| Sol | $59.77_{(1.21)}$ | $57.73_{(1.33)}$ | $70.18_{(0.63)}$ | $70.12_{(0.31)}$ | $64.43_{(0.25)}$ | $67.33_{(1.46)}$ | $68.15_{(0.92)}$ | $59.17_{(0.21)}$ | $70.23_{(0.75)}$ | $67.02_{(0.40)}$ | 77.0 (DeepSol [39]) |
| **Localization Prediction** | | | | | | | | | | | |
| Sub | $49.17_{(0.40)}$ | $31.13_{(0.47)}$ | $62.98_{(0.37)}$ | $56.02_{(0.82)}$ | $58.73_{(1.05)}$ | $52.30_{(3.51)}$ | $76.53_{(0.93)}$ | $59.44_{(0.16)}$ | $78.13_{(0.49)}$ | $79.82_{(0.18)}$ | 86.0 (LA-ProtT5 [79]) |
| Bin | $77.43_{(0.42)}$ | $55.63_{(0.85)}$ | $88.11_{(0.14)}$ | $75.74_{(0.74)}$ | $82.67_{(0.32)}$ | $78.99_{(4.41)}$ | $91.32_{(0.89)}$ | $81.54_{(0.09)}$ | $92.40_{(0.35)}$ | $91.61_{(0.10)}$ | 92.34 (DeepLoc [2]) |
| **Structure Prediction** | | | | | | | | | | | |
| Cont | - | - | $26.34_{(0.65)}$ | $17.50_{(0.77)}$ | $10.00_{(0.20)}$ | $20.43_{(0.74)}$ | $39.66_{(1.21)}$ | $24.35_{(0.44)}$ | $45.78_{(2.73)}$ | $40.37_{(0.22)}$ | 82.1 (MSA Transformer [64]) |
| Fold | $9.57_{(0.46)}$ | $7.10_{(0.56)}$ | $8.24_{(1.61)}$ | $8.52_{(0.63)}$ | $10.93_{(0.35)}$ | $8.89_{(1.45)}$ | $16.94_{(0.42)}$ | $10.74_{(0.93)}$ | $28.17_{(2.05)}$ | $29.95_{(0.21)}$ | 56.5 (GearNet-Edge [104]) |
| SSP | - | - | $68.99_{(0.76)}$ | $59.62_{(0.94)}$ | $66.07_{(0.06)}$ | $69.56_{(0.20)}$ | $82.18_{(0.05)}$ | $62.51_{(0.06)}$ | $82.73_{(0.21)}$ | $83.14_{(0.10)}$ | 86.41 (DML_SS$^{embed}$ [100]) |
| **Protein-Protein Interaction Prediction** | | | | | | | | | | | |
| Yst | $55.83_{(3.13)}$ | $53.00_{(0.50)}$ | $53.62_{(2.72)}$ | $54.12_{(1.27)}$ | $55.07_{(0.02)}$ | $48.91_{(1.78)}$ | $63.72_{(2.80)}$ | $53.87_{(0.38)}$ | $57.00_{(6.38)}$ | $66.07_{(0.58)}$ | - |
| Hum | $62.77_{(2.30)}$ | $54.67_{(4.43)}$ | $63.75_{(5.12)}$ | $59.58_{(2.09)}$ | $62.60_{(1.67)}$ | $68.61_{(3.78)}$ | $77.32_{(1.10)}$ | $83.61_{(1.34)}$ | $78.17_{(2.91)}$ | $88.06_{(0.24)}$ | - |
| Aff | $2.908_{(0.043)}$ | $2.984_{(0.026)}$ | $2.853_{(0.124)}$ | $2.499_{(0.156)}$ | $2.796_{(0.071)}$ | $3.005_{(0.244)}$ | $2.195_{(0.073)}$ | $2.996_{(0.462)}$ | $2.281_{(0.250)}$ | $3.031_{(0.014)}$ | - |
| **Protein-Ligand Interaction Prediction** | | | | | | | | | | | |
| PDB | - | - | $1.457_{(0.131)}$ | $1.455_{(0.070)}$ | $1.376_{(0.008)}$ | $1.441_{(0.064)}$ | $1.562_{(0.072)}$ | $1.457_{(0.024)}$ | $1.559_{(0.164)}$ | $1.368_{(0.076)}$ | 1.181 (SS-GNN [103]) |
| BDB | - | - | $1.572_{(0.022)}$ | $1.566_{(0.052)}$ | $1.497_{(0.022)}$ | $1.565_{(0.033)}$ | $1.549_{(0.019)}$ | $1.649_{(0.022)}$ | $1.556_{(0.047)}$ | $1.571_{(0.032)}$ | 1.34 (DeepAffinity [37]) |

* Used as a feature extractor with pre-trained weights frozen.

previous findings in the literature [27, 83, 78] that dipeptide statistical features are informative for protein understanding.

- **Among models trained from scratch, shallow CNN is the best model.** Among four protein sequence encoders trained from scratch, the shallow CNN performs best on 8 out of 17 tasks and achieves competitive performance on others. These results align with the finding in Shanehsazzadeh *et al.* [74] that, when trained from scratch, a shallow CNN model is no worse or even better than deeper models like LSTM, Transformer and ResNet.

- **ESM-1b is a superior model for various benchmarks.** ESM-1b obtains the best performance on 13 out of 17 benchmark tasks, and it performs consistently well either as a feature extractor or fine-tuned along with the predictor. These results verify ESM-1b as a superior protein language model capturing rich evolutionary and biological patterns of protein sequences. Comprehending such effectiveness is important for model explainability. We leave it as a major future work.

### 5.3 Benchmark Results on Multi-Task Learning

In Tab. 4, we report the benchmark results for multi-task learning. Following the principle that "protein structures determine their functions" [29], we employ three structure prediction tasks, *i.e.*, contact prediction, fold classification and secondary structure prediction, as the auxiliary task, and we evaluate the effect of training each of these tasks along with the center task. We perform multi-task learning on three models with different capacities, *i.e.*, shallow CNN, Transformer and ESM-1b. According to experimental results, we have following findings:

- **MTL well benefits shallow CNN.** By using contact prediction or secondary structure prediction as the auxiliary task, the performance of CNN is improved by 7.10% and 2.45% relative to single-task learning, respectively. Therefore, these two tasks can be deemed as the suitable auxiliary task for CNN. By comparison, the MTL assisted by fold classification is less benefical to CNN, which leads to 2.10% relative performance decay.

- **MTL least benefits the Transformer trained from scratch.** Compared to CNN and ESM-1b, the Transformer model achieves least performance gain from MTL. In fact, compared to single-task learning, relative performance decay is brought by all three considered auxiliary tasks.

- **MTL consistently benefits ESM-1b.** The ESM-1b model is broadly benefited by applying all the three auxiliary tasks. In particular, contact prediction improves the single-task learning baseline by 3.72% on average. Fold classification and secondary structure prediction also bring the average relative improvements of 1.01% and 1.70%, respectively.

Table 4: Benchmark results on multi-task learning. We report *mean (std)* for each experiment. Red results outperform the *original* single-task learning baseline; gray results are same as the baseline; blue results underperform the baseline; **bold results** denote best improvements achieved on the center task or brought by the auxiliary task; "-" indicates not applicable for this setting. *Abbr.*, Ori.: original; $\overline{\text{Rel.}}$ ↑/↓: average relative improvement/decay.

| Task | CNN | | | | | Transformer | | | | | ESM-1b | | | | |
|---|---|---|---|---|---|---|---|---|---|---|---|---|---|---|---|
| | Ori. | +Cont | +Fold | +SSP | $\overline{\text{Rel.}}$ ↑/↓ | Ori. | +Cont | +Fold | +SSP | $\overline{\text{Rel.}}$ ↑/↓ | Ori. | +Cont | +Fold | +SSP | $\overline{\text{Rel.}}$ ↑/↓ |
| **Function Prediction** | | | | | | | | | | | | | | | |
| GB1 | 0.502(0.007) | 0.692(0.091) | 0.507(0.012) | 0.548(0.005) | ↑16.00% | 0.271(0.020) | 0.386(0.034) | 0.391(0.090) | 0.289(0.031) | ↑**31.12%** | 0.705(0.019) | 0.694(0.025) | 0.710(0.024) | 0.709(0.061) | ↓0.09% |
| AAV | 0.746(0.003) | 0.752(0.043) | 0.772(0.008) | 0.791(0.004) | ↑3.44% | 0.681(0.013) | 0.730(0.001) | 0.699(0.018) | 0.717(0.023) | ↑5.04% | 0.821(0.010) | 0.797(0.019) | 0.799(0.037) | 0.825(0.011) | ↓1.71% |
| Thermo | 0.494(0.021) | 0.537(0.016) | 0.561(0.002) | 0.558(0.007) | ↑11.74% | 0.545(0.031) | 0.561(0.009) | 0.412(0.001) | 0.414(0.010) | ↓15.17% | 0.669(0.028) | 0.668(0.006) | 0.661(0.015) | 0.671(0.002) | ↓0.35% |
| Flu | 0.682(0.002) | 0.680(0.001) | 0.682(0.001) | 0.683(0.001) | ↓0.05% | 0.643(0.005) | 0.642(0.017) | 0.648(0.000) | 0.656(0.002) | ↑0.88% | 0.678(0.001) | 0.681(0.001) | 0.679(0.001) | 0.681(0.002) | ↑0.34% |
| Sta | 0.637(0.010) | 0.661(0.006) | 0.472(0.170) | 0.695(0.016) | ↓4.34% | 0.649(0.056) | 0.620(0.004) | 0.672(0.010) | 0.667(0.063) | ↑0.62% | 0.694(0.073) | 0.733(0.007) | 0.728(0.002) | 0.759(0.002) | ↑6.63% |
| β-lac | 0.781(0.011) | 0.835(0.009) | 0.736(0.012) | 0.811(0.014) | ↑1.66% | 0.261(0.015) | 0.142(0.063) | 0.276(0.029) | 0.197(0.017) | ↑21.46% | 0.839(0.053) | 0.899(0.001) | 0.882(0.007) | 0.881(0.001) | ↑5.76% |
| Sol | 64.43(0.25) | 70.63(0.34) | 69.23(0.10) | 69.85(0.62) | ↑8.50% | 70.12(0.31) | 70.03(0.42) | 68.85(0.43) | 69.81(0.46) | ↓0.78% | 70.23(0.75) | 70.46(0.16) | 64.80(0.49) | 70.03(0.15) | ↓2.56% |
| **Localization Prediction** | | | | | | | | | | | | | | | |
| Sub | 58.73(1.05) | 59.07(0.45) | 56.54(0.65) | 56.64(0.33) | ↓2.24% | 56.01(0.81) | 52.92(0.64) | 56.74(0.29) | 56.70(0.16) | ↓0.99% | 78.13(0.49) | 78.86(0.75) | 78.43(0.28) | 78.00(0.34) | ↑0.38% |
| Bin | 82.67(0.32) | 82.67(0.72) | 81.14(0.40) | 81.83(0.86) | ↓0.96% | 75.74(0.74) | 74.98(0.77) | 76.27(0.57) | 75.20(1.23) | ↓0.34% | 92.40(0.34) | 92.50(0.26) | 91.83(0.20) | 92.26(0.20) | ↓0.22% |
| **Structure Prediction** | | | | | | | | | | | | | | | |
| Cont | 10.00(0.20) | - | 5.87(0.21) | 5.73(0.66) | ↓42.00% | 17.50(0.77) | - | 2.04(0.31) | 12.76(1.62) | ↓57.71% | 45.78(2.72) | - | 35.86(1.27) | 32.03(12.2) | ↓25.85% |
| Fold | 10.93(0.35) | 11.07(0.38) | - | 11.67(0.56) | ↑4.03% | 8.62(0.62) | 9.16(0.91) | - | 8.14(0.76) | ↑0.35% | 28.10(2.05) | 32.10(0.72) | - | 28.63(1.55) | ↑8.06% |
| SSP | 66.07(0.06) | 66.13(0.06) | 65.93(0.04) | - | ↓0.06% | 59.62(0.94) | 63.10(0.43) | 50.93(0.26) | - | ↓4.37% | 82.73(0.20) | 83.21(0.32) | 82.27(0.23) | - | ↑0.01% |
| **Protein-Protein Interaction Prediction** | | | | | | | | | | | | | | | |
| Yst | 55.07(1.68) | 54.50(1.61) | 53.28(1.91) | 54.12(2.87) | ↓2.00% | 54.12(1.26) | 52.86(1.15) | 54.00(2.58) | 54.00(1.17) | ↓0.92% | 57.00(6.37) | 58.50(2.15) | 64.76(1.42) | 62.06(5.98) | ↑8.37% |
| Hum | 62.60(1.67) | 65.10(2.26) | 69.03(2.68) | 66.39(0.86) | ↑6.77% | 59.58(2.08) | 60.76(6.67) | 54.80(2.06) | 54.00(2.06) | ↑2.32% | 78.16(2.90) | 81.66(2.88) | 80.28(1.27) | 83.00(0.88) | ↑4.46% |
| Aff | 2.796(0.071) | 1.732(0.044) | 2.392(0.041) | 2.270(0.041) | ↑**23.77%** | 2.499(0.156) | 2.733(0.126) | 2.524(0.146) | 2.651(0.034) | ↓5.48% | 2.280(0.249) | 1.893(0.064) | 2.002(0.065) | 2.031(0.031) | ↑**13.36%** |
| **Protein-Ligand Interaction Prediction** | | | | | | | | | | | | | | | |
| PDB | 1.376(0.008) | 1.328(0.033) | 1.316(0.064) | 1.295(0.030) | ↑4.58% | 1.455(0.069) | 1.574(0.215) | 1.531(0.181) | 1.387(0.019) | ↓2.91% | 1.559(0.164) | 1.458(0.003) | 1.435(0.015) | 1.419(0.026) | ↑7.80% |
| BDB | 1.497(0.022) | 1.501(0.035) | 1.462(0.044) | 1.481(0.036) | ↑1.05% | 1.566(0.051) | 1.490(0.058) | 1.464(0.007) | 1.519(0.050) | ↑4.79% | 1.556(0.047) | 1.490(0.033) | 1.511(0.017) | 1.482(0.014) | ↑3.96% |
| $\overline{\text{Rel.}}$ ↑/↓ | - | ↑**7.10%** | ↓2.10% | ↑2.45% | - | - | ↓0.33% | ↑3.57% | ↓2.05% | - | - | ↑3.72% | ↑1.01% | ↑1.70% | - |

- **Contact prediction is easily dominated by auxiliary task.** Under all three considered models, the performance of contact prediction degenerates significantly when trained along with the auxiliary task. This phenomenon indicates that the training objective of contact prediction is more vulnerable than those of other benchmark tasks.

In summary, MTL has great potential on boosting the performance of CNN and ESM-1b, while negative effects show on the Transformer trained from scratch. We thus suggest CNN and ESM-1b as baseline models for future MTL research. In addition, suitable auxiliary tasks could vary across models, requiring an extra auxiliary task selection scheme. We suggest it as a future direction.

## 6    Conclusions and Future Work

In this work, we build a comprehensive benchmark for general protein sequence understanding, named as PEER. Upon this benchmark, we compare the performance of both single- and multi-task learning among three types of models. The benchmark results for single-task learning demonstrate the shallow CNN as a decent baseline model which owns competitive performance and low computational cost, and ESM-1b is verified as a superior protein language model. The benchmark results for multi-task learning demonstrate the effectiveness of MTL on boosting the performance of CNN and ESM-1b, and they also illustrate the importance of selecting suitable auxiliary tasks.

In the future, we will extend the current benchmark by going beyond sequence-based datasets and methods to structure-based ones. Also, we will continue promoting the efforts on MTL for protein sequence understanding, and we would like to collaborate with the community to push the edge of this research topic.

## Acknowledgments

This project is supported by AIHN IBM-MILA partnership program, the Natural Sciences and Engineering Research Council (NSERC) Discovery Grant, the Canada CIFAR AI Chair Program, collaboration grants between Microsoft Research and Mila, Samsung Electronics Co., Ltd., Amazon Faculty Research Award, Tencent AI Lab Rhino-Bird Gift Fund, a NRC Collaborative R&D Project (AI4D-CORE-06) as well as the IVADO Fundamental Research Project grant PRF-2019-3583139727.

The authors would like to thank Meng Qu, Shengchao Liu, Chence Shi, Minkai Xu, Huiyu Cai and Hannes Stärk for their helpful discussions and comments.

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
