# OpenReview forum: "PEER: A Comprehensive and Multi-Task Benchmark for Protein Sequence Understanding"
_NeurIPS.cc/2022/Track/Datasets_and_Benchmarks — NeurIPS 2022 Datasets and Benchmarks _

### Official Review · Reviewer_17gX · 2022-07-20
**A very excellent and useful benchmark providing much convenience for researchers interested in protein sequence representation learning**

**Rating:** 8
**Confidence:** 4
**Clarity:** The writing of this paper is good and…

**Strengths:**

(+) This work proposes the PEER benchmark datasets, covering more comprehensive protein sequence representation tasks and data than previous benchmark datasets. The creation of new and large benchmark will provide chances for researchers to more comprehensively and systematically evaluate their proposed new methods.
(+) The implementations of multiple popular protein sequence models are provided in the proposed benchmark, making it more convenient to reproduce or compare with the results of baseline methods.
(+) This work proposes a novel and insightful idea of improving protein sequence models by multi-task learning, which is very useful and enlightening for researchers in the protein representation learning area.

**Weaknesses:**

(-) Currently, the PEER benchmark focuses on protein sequence representation learning, but some recent methods have been proposed to do protein structure representation learning. I hope the PEER benchmark can offer 3D structures of proteins in the dataset if possible, and even include some structure representation baseline methods in the future.

**Additional Feedback:**

N/A

**Correctness:**

Both the dataset collection and baseline evaluation are sound in this work. The experiments are appropriately designed and all claims are supported by solid experiment results.

**Documentation:**

Detailed information about the dataset creation, code implementation, reproducing experimental results, and future open source and maintenance plans are provided in the supplementary file. In addition, a URL for access to the dataset has been provided in authors' private comments.

**Ethics:**

No ethical review is needed.

**Relation To Prior Work:**

This paper clearly discussed the difference between the proposed benchmark and previous benchmarks.

**Summary And Contributions:**

This work proposes PEER, a benchmark for protein sequence representation learning, including curated datasets and implemented baseline methods. Comprehensive experiments are conducted to evaluate the performance of multiple protein sequence models and multi-task learning.

---

> ### Author Response · Authors · 2022-08-09
> **Author Feedbacks to Reviewer 17gX**
>
> >**Q1: I hope the PEER benchmark can offer 3D structures of proteins in the dataset if possible, and even include some structure representation baseline methods in the future.**
>
> Thanks for this excellent point. This is exactly what we plan to do in the near future!
>
> Accompanied with PEER benchmarks, the code of another work from our group [a] will be released in the [TorchProtein library](https://torchprotein.ai/). [a] focuses on pre-training with 3D protein structures and includes both sequence- and structure-based baseline methods. In the future, we will include 3D structures for datasets in PEER and all tasks in Atom3D [b] and implement existing structure-based encoders for proteins, e.g., GearNet/GearNet-Edge [a], IEConv [c], GVP [d] and ProNet [e]. Since the aim of the PEER benchmark is to test the capacity of sequence-based models, we’d like to leave this 3D benchmark as future work.
>
> [a] Protein representation learning by geometric structure pretraining. Zhang et al, arXiv preprint  arXiv:2203.06125, 2022.
>
> [b] Atom3d: Tasks on molecules in three dimensions. Townshend et al. arXiv preprint arXiv:2012.0403, 2020.
>
> [c] Intrinsic-extrinsic convolution and pooling for learning on 3d protein structures. Hermosilla et al., ICLR, 2021.
>
> [d] Learning from protein structure with geometric vector perceptrons. Jing et al. ICLR, 2021.
>
> [e] Learning Protein Representations via Complete 3D Graph Networks. Wang et al. arXiv preprint arXiv:2207.12600, 2022.

---

### Official Review · Reviewer_dcmR · 2022-07-20
**A benchmark for protein sequence understandings with limited research contributions**

**Rating:** 4
**Confidence:** 5
**Correctness:** This part is fine
**Clarity:** yes

**Strengths:**

(1)A benchmark covers many tasks
(2)relevance to the compuational biology community

**Weaknesses:**

(1) most tasks, datasets and baselines were already existed in literature
(2) contact prediction and secondary structure prediction tasks are no longer interesting given the remarkable results in 3D structure prediction by alphaFold2
(3) tasks such as PPI  prediction is not interesting from biology perspective.
(4)A lack of more advanced baselines such as MSA transformer and Evoformer.

**Additional Feedback:**

no

**Documentation:**

in general fine

**Relation To Prior Work:**

They have presented most relevant literature but such papers are too much in literature.

**Summary And Contributions:**

This paper is well written and understandable. The authors present a comprehensive benchmark for several protein sequence based prediction tasks, such as fitness prediction, ppi prediction, structural predictions. As for baselines, they choose some typical NLP models, such as LSTM  CNN and Transformer. In general this paper has limited novelty and contributions：

First, most of thest tasks the paper present have already had well-established benchmarks. For example, for these fitness prediction tasks, they have occured in TAPE （Raoshan2019） and FLIP （Yang 2021）bechmarks. I did not see the necessarities introducing this benchmark again. Another example is structure prediction, there are also various beanchmarks, such as TAPE again. More importantly. the proposed structure prediction tasks are not important to the community now given that AlphaFold2 can predict extremely high accuracy for 3D structure, why we still need secondary and contact predict tasks?   Second, for these PPI prediction task, I would say for most biologists they would not care the accuracy of such binary prediction tasks.  They care more about which part or which postions there are interactions, particularly from 3D structure perspective, not a binary value yes or no. This paper is very AI-style without deeply considering  the key challenges in biology field.

In addition, they present only four types of baselines, LSTM transformer, cnn and resnet. What about some typical compuational baselines  such as Gremlin and more advanced baselines  such MSA transformer and Evoformer (the language model of AlphaFold).

In summary, I think this paper is not good enough for NeuralPS. I suggest rejection.

---

> ### Author Response · Authors · 2022-08-09
> **Author Feedbacks to Reviewer dcmR (Part 1/3)**
>
> >**Q1: Most tasks, datasets and baselines already existed in literature.**
>
> We agree on the existence of most benchmark tasks, datasets and baseline models. However, we argue that, **by combining all the tasks, datasets, baseline models and the novel multi-task learning setting, our PEER benchmark is of great value in the community of protein machine learning.** Detailed points are as follows:
>
> 1. For a long time, researchers in this community have to adapt their custom models to the codebase of FLIP/TAPE when they are to deal with function prediction, to the codebase of DeepLoc when they are to deal with localization prediction, and to the codebase of TAPE when they are to deal with several subproblems of structure prediction. There is no guarantee that the adaptations performed by different researchers are rigorously consistent, facing the risks of unfair comparisons. At the core of applied machine learning, rigorously fair benchmarks facilitate the fair comparison among different models and thus push the edge of various application domains, definitely including protein science. Under such context, our benchmark shows its unique value of **fairly evaluating the general effectiveness of a model on a diverse suite of protein understanding tasks** covering protein function/localization/structure prediction, PPI prediction and PLI prediction.
> 2. Furthermore, we would like to point out another important contribution of PEER that **diverse types of models are evaluated under strictly aligned experimental configurations on all tasks**. To be specific, PEER evaluates 10 models within three model types (feature engineers, protein sequence encoders and protein language models) on each benchmark task, with identical training configurations (training epoch, optimizer, loss function, etc.) and identical validation-test configurations (model selection metric, validation interval, test metric, etc.). By comparison, TAPE does not cover the evaluation of feature engineers and up-to-date protein language models; FLIP does not cover the evaluation of feature engineers and multiple protein sequence encoders; and both benchmarks do not provide detailed experimental configurations for strict reproduction of results. In our source code release, we will release all the dataset, task, model, training and evaluation implementations and all detailed experimental configurations (separate yaml format files for each experiment) to the community, so as to promote reproducibility and fair comparisons in this fast-growing field.
> 3. Last but not least, based on multiple tasks and task types in the PEER benchmark, it further **enables the study of Multi-Task Learning**, e.g., how to boost a model’s performance on a function prediction task by jointly learning a structure prediction task (our benchmark experiments have an attempt in this way). This new research line has great potential to improve the generalization ability and versatility of protein sequence understanding models.
>
> Therefore, considering all the contributions as a comprehensive benchmark for fair model comparison, the best practices of reproducible benchmark research and a multi-task benchmark for multi-task learning research, the PEER benchmark is uniquely valuable in this field.
>
>
> >**Q2: Contact prediction and secondary structure prediction tasks are no longer interesting given the remarkable results in 3D structure prediction by AlphaFold2.**
>
> We respectfully disagree with the reviewer on this point. The focus of our benchmark is to **evaluate the effectiveness of various protein sequence representation learning methods, instead of identifying the SOTA model on protein folded structure prediction.** Contact prediction and secondary structure prediction are **suitable benchmark tasks for evaluating protein representation learning** that (1) define important subproblems of protein structure prediction, (2) are widely adopted in the community for benchmarking purposes, and (3) can be handled by a broad variety of models, which facilitates the diversity of baseline models.
>
> In addition, we would like to point out that **studying on these relatively easy structure prediction subproblems can also facilitate the development of their ultimate goal, protein folded structure prediction.** Specifically, by solving contact prediction and secondary structure prediction, we can identify superior protein language models with promising performance. These identified language models can be further injected into a SOTA protein folded structure predictor (e.g., AlphaFold2) and seek to improve its performance.

---

> > ### Author Response · Authors · 2022-08-09
> > **Author Feedbacks to Reviewer dcmR (Part 2/3)**
> >
> > >**Q3: Tasks such as PPI prediction are not interesting from a biology perspective.**
> >
> > We respectfully disagree with the reviewer on this point. (1) For the PPI affinity prediction task, given a target protein, it seeks to predict the relative binding strength among candidate binders, which is **important in the applications of protein binder design**, just as done in two recent projects [a,b]. Also, this benchmark task splits the dataset based on a multi-round protein binder design scenario, i.e., putting lower-order mutants in the training set and putting higher-order mutants in the validation and test sets. Therefore, it provides a test field for machine learning models in a simulated real-world application. (2) For the Human PPI prediction task, it seeks to predict whether two human proteins interact or not, which is **a basic goal of constructing the human protein interactome**. Unraveling the human protein interactome is vital to understand mechanisms of disease and uncover unknown disease genes, motivating many previous [c,d] and ongoing [e,f] projects. This benchmark task is expected to contribute by boosting effective and high-throughput machine learning models for human PPI prediction. (3) For the Yeast PPI prediction task, it aims at predicting whether two yeast proteins interact ot not, which is **a basic goal of constructing the yeast protein interactome**. It is of broad scientific interest to construct complete and precise yeast interactome network maps, motivating many previous [g,h] and ongoing [i] projects. This benchmark task will aid to these projects by predicting binary yeast protein interactions with effective and high-throughput machine learning models.
> >
> > We have supplemented these impacts of PPI prediction tasks in the Section 3.4 of the revised main paper.
> >
> > [a] Deep geometric representations for modeling effects of mutations on protein-protein binding affinity. Liu et al., PLoS computational biology, 2021.
> >
> > [b] Deep learning guided optimization of human antibody against SARS-CoV-2 variants with broad neutralization. Shan et al., Proceedings of the National Academy of Sciences, 2022.
> >
> > [c] Next-generation sequencing to generate interactome datasets. Yu et al., Nature methods, 2011.
> >
> > [d] A proteome-scale map of the human interactome network. Rolland et al., Cell, 2014.
> >
> > [e] OpenPIP: An Open-source Platform for Hosting, Visualizing and Analyzing Protein Interaction Data." Helmy et al., Journal of Molecular Biology, 2022. [Project Website](http://openpip.baderlab.org/)
> >
> > [f] The Human Reference Protein Interactome Mapping Project. [Vidal](http://ccsb.dfci.harvard.edu/web/www/ccsb/), [Roth](http://llama.mshri.on.ca/), [Tavernier](http://www.vib.be/en/research/scientists/Pages/Jan-Tavernier-Lab.aspx), and [Bader](http://www.baderlab.org) labs. [Project Website](http://www.interactome-atlas.org/)
> >
> > [g] High-quality binary protein interaction map of the yeast interactome network. Yu et al., Science, 2008.
> >
> > [h] Quantitative analysis of fitness and genetic interactions in yeast on a genome scale. Baryshnikova et al., Nature methods, 2010.
> >
> > [i] The Yeast Protein Interactome Mapping Project. [Vidal](http://ccsb.dfci.harvard.edu/web/www/ccsb/), [Roth](http://llama.mshri.on.ca/), [Tavernier](http://www.vib.be/en/research/scientists/Pages/Jan-Tavernier-Lab.aspx), and [Bader](http://www.baderlab.org) labs. [Project Website](http://yeast.interactome-atlas.org/)

---

> > > ### Author Response · Authors · 2022-08-09
> > > **Author Feedbacks to Reviewer dcmR (Part 3/3)**
> > >
> > > >**Q4: A lack of more advanced baselines such as MSA transformer and Evoformer.**
> > >
> > > **Table A. Performance comparison on fold classification. We report the mean (std) of test accuracy over three runs with seeds 0, 1 and 2.**
> > >
> > > |Model|ProtBert|ESM-1b|MSA-Transformer|
> > > |:----:|:----:|:----:|:----:|
> > > |Fold classification (Acc)|16.94 (0.42)|28.17 (2.05)|17.46 (1.08)|
> > >
> > > Thanks for pointing out these important advanced baselines. Both MSA Transformer and Evoformer rely on MSAs as inputs. Because of the high time cost of retrieving MSA data for our benchmark tasks. During this response period, we select the fold classification task for MSA retrieval and have finished the evaluation on MSA Transformer. In Table A, we present the performance comparison between MSA Transformer, ProtBert and ESM-1b on fold classification. Because of the high memory cost of fine-tuning MSA Transformer, we can only set the MSA depth (i.e., the number of homologous sequences used for each original sequence) as 8 and the batch size as 2 to fully fit the memory of the Tesla V100 (32GB) GPU.
> > >
> > > We can observe that the MSA Transformer outperforms ProtBert while lags behind ESM-1b. We believe the performance of MSA Transformer can be further enhanced by using larger MSA depth, while, to achieve this, we have to address the expensive computation of fine-tuning MSA Transformer. We will continue working on this and also the evaluation of Evoformer in the following weeks. Also, we will try our best to retrieve MSAs for more benchmark tasks for evaluation. All these data, models and results will be added to our project and paper in the near future.

---

> > > > ### Comment · Reviewer_dcmR · 2022-08-23
> > > > **thanks for your response.**
> > > >
> > > > Thanks for your point by point reply.
> > > >
> > > > The reason I reject this paper is mainly because after going through it I did not find much new materials (neither experiments nor key points). Most of these tasks are widely existed in literature. Both datasets, codes and  benchmarks are quite available in some very old literature. It will not be interesting to accept another paper doing similar things.  For NeuralPS, I would like to see the paper make some significant contributions to the community or at least show some new insights. These tasks present in this paper are too old and some tasks like SSP and contact prediction (appear in much past benchmark literature) are not that interested from the biologist perspectives given the breakthrough made by AlphaFold (Note you are presenting a benchmark to solve biological challenges, if it is not a challenge, why you still present it and make the AI community to contribute it; Second, in TAPE and ESM1b, there are already released datasets and code.).  These baselines introduced (LSTM ResNet CNN Transformer) do not have any biological insights. I know they appeared in some past literature but these baselines were proposed around 2019.  I guess I may give a weak accept if this paper was submitted in 2019 or before. Overall, I think the current version is under the bar of NeuralPS.  I would encourage these authors to explore biological problems in more depth --- solving some issues by combining important biological insights.

---

> > > > > ### Author Response · Authors · 2022-08-23
> > > > > **Follow-up Feedbacks to Reviewer dcmR**
> > > > >
> > > > > Dear Review dcmR,
> > > > >
> > > > > Thanks so much for your response. We appreciate your points on benchmark task selection from the perspective of biological importance.
> > > > >
> > > > > We would like to argue the contributions of our work from following two points:
> > > > >
> > > > > (1) **(Done by our benchmark) Our comprehensive PEER benchmark is a unique test field for superior and versatile protein language models. (Leave for practitioners) The top candidates after benchmarking can be applied to the tasks interested by biologists.**  Given the quick rise of protein language models (PLMs) trained by different groups [a,b,c,d,e,f], it is important to have a collective benchmark to identify the top candidates on a broad and diverse suite of benchmark tasks. Our PEER benchmark will serve this unique role in the community, *i.e.*, setting up a suite of 17 tasks lying in 5 task groups and **continuously evaluating the newest PLMs released to the community** (our next-step evaluation includes ESM2 [a], MSA Transformer [b], Evoformer [c], *etc.*). **All these efforts will be actively updated on our [project website](https://torchprotein.ai/benchmark).**
> > > > >
> > > > > Based on these benchmark results, biologists will be aware of the newest PLMs and their performance against previous ones. The leaderboards for all tasks or a specific task category (*e.g.*, structure/function prediction) will further guide them to apply leading PLMs to their specific tasks of biological interests. For protein structure prediction, since ESMFold [a] has observed a high correlation between structure prediction performance and the quality of PLM representations, the leading PLMs identified by our benchmark are expected to help more accurate structure prediction. For protein engineering, PLMs with informative representations are demonstrated to reveal evolutionary patterns for better protein functions [g,h], and thus the leading PLMs in our benchmark are strong candidates for protein engineering applications. We will make sure that our suggestions of leading PLMs keep up with the latest progress, so that biological practitioners will have the sense of choosing a decent PLM to boost their projects.
> > > > >
> > > > > (2) **The PEER benchmark enables the study of Multi-Task Learning.** Our benchmark integrates five types of protein understanding tasks, ranging from function/localization/structure prediction to protein-protein and protein-ligand interaction prediction, which opens the opportunity for multi-task learning research, *e.g.*, how to boost a model’s performance on a function prediction task by jointly learning a structure prediction task (the experiments in our manuscript have an attempt along this way). These preliminary efforts are expected to inspire the community on studying how to **improve the generalization ability and versatility of protein sequence understanding models by learning multiple tasks together**.
> > > > >
> > > > > Based on these two points, we believe the unique value of our project in this field, *i.e.*, **a long-term project of identifying up-to-date superior PLMs** and **a pioneering project towards multi-task learning for protein understanding**. We promise to keep it growing and thriving along with the whole community.
> > > > >
> > > > > &emsp;
> > > > >
> > > > > Best regards,
> > > > >
> > > > > Authors of Paper #64
> > > > >
> > > > > &emsp;
> > > > >
> > > > > [a] Language models of protein sequences at the scale of evolution enable accurate structure prediction. Lin et al., bioRxiv, 2022.
> > > > >
> > > > > [b] MSA Transformer. Rao et al., ICML, 2021.
> > > > >
> > > > > [c] Highly accurate protein structure prediction with AlphaFold. Jumper et al., Nature, 2021.
> > > > >
> > > > > [d] Convolutions are competitive with transformers for protein sequence pretraining. Yang et al., bioRxiv, 2022.
> > > > >
> > > > > [e] RITA: a Study on Scaling Up Generative Protein Sequence Models. Hesslow et al., arXiv, 2022.
> > > > >
> > > > > [f] Tranception: protein fitness prediction with autoregressive transformers and inference-time retrieval. Notin et al., ICML, 2022.
> > > > >
> > > > > [g] Learning the language of viral evolution and escape. Hie et al., Science, 2021.
> > > > >
> > > > > [h] Learning mutational semantics. Hie et al., NeurIPS, 2020.

---

### Official Review · Reviewer_angS · 2022-07-27
**A much needed benchmark for fast-tracking protein understanding**

**Rating:** 7
**Confidence:** 3
**Correctness:** Looks good to me.

**Strengths:**

Great diversity in benchmark, in models, in training setup, etc. Benchmark is much needed in the field.

Haven't looked at the code in details, but looks good at first glance.

**Weaknesses:**

The presentation is too cluttered. Difficult to gather information quickly from the text and the tables. Things can be structured and summarized better using the right structures.

Some information about the datasets is missing such as a number of unique proteins, the split used, the protein lengths, etc.

Some more studies could be done to understand the importance of language models.

**Additional Feedback:**

See above comments.

**Clarity:**

Could be better. Too cluttered and difficult to find the information needed.

That being said, writing is okay.

**Documentation:**

Not sure.

**Ethics:**

Looks good.

**Relation To Prior Work:**

Looks good to me.

**Summary And Contributions:**

PEER proposes a Benchmark for protein models on a variety of tasks, including protein function, localisation, structure, PPI, and ligand affinity. It evaluates a variety of methods, including feature engineering, deep learning (CNN, LSTM, Transformer, ResNet), pre-trained protein models (frozen VS unfrozen).

It is a much-needed paper in the community and will help advance the field of protein understanding. It brings a novel contributions on many aspects. However, the paper is a bit cluttered and it is difficult to get a global picture of what's going on and what are the different tasks. What does the paper need? Better Tables! And more fluidity in the text to know where it is going, not a simple enumeration.

1. Sections 4.1.1, 4.1.2 and 4.1.3 can be summarized in a single table with only the most important info to compare the architectures.

2. Table 1 is missing many information. Perhaps flip the page into landscape mode and do a full-page table?

    a. The acronym to correspond the entries of Table 1 with Table 2.

    b. A short description of the task

    c. The number of unique proteins

    d. Average and std of the protein sequence length

    e. The train/val/test split method

    f. Other info specific to the dataset: Number of unique ligands for for BindingDB. Mean and std for the regression tasks. The data imbalance for the classification tasks.

3. The color in Table 2 is not informative and makes it very hard to read. Some values can be very close to each other, yet the strong color difference suggest that they are far. Instead, I would highly suggest simply doing a color-scale or "conditional formatting" for each row. The best in dark, the worst in white. That way, one can easily find which column is usually the best method. Relative comparison within a group will still be possible. Also, I suggest using a color such as green instead of gray to make it easier to read the table.

4. Table 3 could add a row at the bottom of average relative improvement, and another column at the right of each group for the same relative improvement. For example, +10% indicates that, in average, metrics improve by a relative 10% compared to the original. Again, this is more informative than a single blue/red color, and allow to see which task benefits the most, and which model benefits the most.

5. Analysis of the correlation between performance VS size of the training set VS length of proteins. This should be done for at least one model per category (feature engineer, deep learning, pre-trained models), and will help to see if they play a role in the model's performance.

In general, a good paper and good benchmark! Willing to upgrade my score if concerns are answered since they are mostly about presentation of the results.

---

> ### Author Response · Authors · 2022-08-09
> **Author Feedbacks to Reviewer angS**
>
> >**Q1: Sections 4.1.1, 4.1.2 and 4.1.3 can be summarized in a single table with only the most important info to compare the architectures.**
>
> Thanks for this great advice. In the revised main paper, we summarize the baseline models in Table 2, which describes each model along with its model type, input layer, hidden layers, output layer and number of parameters. We merge the original Sections 4.1.1, 4.1.2 and 4.1.3 into a single Section 4.1 which concisely introduces the high-level ideas of all baseline models.
>
> >**Q2: Table 1 is missing many information.**
>
> Appreciate for these constructive suggestions. In the Table 1 of revision, for each task, we have supplemented its acronym, the number of unique proteins involved in the task, and the mean and std of the protein sequence length. Because of the **allowed-but-not-recommended** attitude from this venue on inserting a landscape-mode page in the main paper, we choose to place **more space-consuming information of each task in texts**, including the dataset splitting method, the general impact of the task (newly added following Reviewer 7t4f’s suggestion) and other task-specific information.
>
> >**Q3: The color in Table 2 is not informative and makes it very hard to read.**
>
> This suggestion is really helpful. In the Table 3 of the revised main paper (corresponding to the original Table 2), we use **four color scales of green** to denote the **first, second, third and fourth best performance among all baseline models**. This visualization clearly shows the superiority of ESM-1b/ESM-1b* (* freezing the pre-trained weights) among all baselines, the best performance of CNN among protein sequence encoders trained from scratch, and the better performance of DDE feature engineer over Moran feature engineer.
>
> Following the suggestion of Reviewer 7t4f, we also add a column at the end of the table to list the SOTA model and its performance from literature for each applicable task. These results help to compare the baseline models in our benchmark with the current best ones in this field.
>
> >**Q4: Table 3 could add a row at the bottom of average relative improvement, and another column at the right of each group for the same relative improvement.**
>
> This suggestion is nice. In Table 4 after revision (corresponding to the original Table 3), we have added a relative improvement column for each model to indicate how each task that the model plays against has benefited from multi-task learning. We also add a relative improvement row at the bottom to indicate how each model is benefited from using a specific auxiliary task. Based on these additions, we have revised the texts in Section 5.3 accordingly.
>
> >**Q5: Analysis of the correlation between performance VS size of the training set VS length of proteins should be done for at least one model per category.**
>
> Thanks for pointing out these important ablation studies. In the Figure 1 of supplementary material, we plot in subfigures (a)&(b) the performance VS truncated sequence length (on subcellular localization prediction and fold classification, respectively) and in subfigure (c) the performance VS training set size (on beta-lactamase activity prediction). In each subfigure, the performance trend of DDE (feature engineer), CNN (protein sequence encoder) and ESM-1b (protein language model) are respectively plotted. For all three models, monotonous performance increase is observed as the increase of truncated sequence length and training set size. These results demonstrate the continuous benefit of having longer protein sequences and more training samples when training a sequence-based protein property prediction model.
>
> For the time limit of this response phase, we have not yet finished the full ablation analysis, i.e., plotting the performance curves against truncated sequence length and against training set size for all five task categories. We are still running these experiments and will update to the next revision when finished.
>
> >**Q6: Some more studies could be done to understand the importance of language models.**
>
> Thanks for the great advice. It is indeed very interesting and important to understand how protein language models capture the structural, biological and evolutionary information of proteins by large-scale pre-training. There are some existing works [a,b] that explore the explainability of protein language models by visualization. We will follow up this direction and study the aspect of explainability in our future works.
>
> [a] BERTology meets biology: interpreting attention in protein language models. Vig et al., arXiv, 2020.
>
> [b] Transformer protein language models are unsupervised structure learners. Rao et al., Biorxiv, 2020.

---

> > ### Comment · Reviewer_angS · 2022-08-10
> > **Thank you for the improvements**
> >
> > Thank you, the paper is much clearer, and it is easier to compare the architectures, tasks and results.
> >
> > I would like to change my score to **7: accept**. The paper will be very useful to the community.

---

> > > ### Author Response · Authors · 2022-08-11
> > > **Thanks for the support! A kind notice that review update is available now :)**
> > >
> > > Dear Reviewer angS,
> > >
> > > Thanks so much for your appreciation and support on our paper and project!
> > >
> > > A recent email from the organizers states that **updating review is available now from the reviewers’ side**, and we have verified this on our own reviewing console. You may like to update the upgraded score on your original review. Thanks again for your support!
> > >
> > > We will continue updating the latest progress of our project to you on the OpenReview platform.
> > >
> > > &emsp;
> > >
> > > Best regards,
> > >
> > > Authors of Paper #64

---

### Official Review · Reviewer_7t4f · 2022-07-27
**Insufficient novelty in benchmark design; counterintuitive MTL results may open interesting research questions with more analyses**

**Rating:** 4
**Confidence:** 4
**Correctness:** N/A

**Strengths:**

The datasets they have chosen in their benchmark cover tasks on different biophysical length-scales and potentially time-scales. A standardized resource to access these multiple dataset types could be convenient for community usage.

The authors also try a panel of different architectures on all tasks, and in tandem. They include feature-engineering models based on biological properties; this is a good inclusion as there are many research groups that use expert-crafted features.

It is interesting to see the MTL results across all 14 datasets, although this was was not done with all 14 tasks simultaneously.

**Weaknesses:**

(1) Several of the function datasets are not generalizable. For example, fluorescence of GFP proteins is not a property relevant to many other protein types. Good performance on this dataset (which can be attained by even lightweight models) may not indicate any knowledge of a broad sequence-to-function relationship. Such criticisms have been brought up before in dedicated fitness benchmarks such as FLIP (Dallago et al.). It is surprising the datasets proposed by FLIP are not included in these tasks.

(2) In general, the metrics for many tasks may not actually reflect a model that learns well. For example, there are notable class imbalances in some of these datasets that can create models that demonstrate strong predictive performance without understanding the sequence-to-output relationship. For example, in the subcellular localization prediction task (DeepLoc), not all classes are equally represented in the data; hence it may be possible to attain a model that is highly accurate on the dominant classes, but unable to distinguish the others. The metric for this task is "accuracy" - something that can be trivially high despite the model truly only "learning" a subset of the classes (if at all). The authors should consider redefining balanced subsets for train/val/test. Alternatively, reporting AUPRC/sensitivity or other metrics that explicitly consider imbalance can alleviate concerns on "high" performance inappropriately reflected.

(3) PLI-type tasks may be just as sensitive to the ligand representation as the sequence representation. How would you account for the protein-sequence contribution in this type of task?

(4) The authors do not clearly state the rationale for choosing these datasets of the many protein datasets that exist. Many of these datasets already exist as benchmarks on their own, but why does this unique set of 14 tasks confer the appropriate way to explore the representational strength of protein modeling?

(5) The MTL results are overstated; improvements, when present, are quite marginal over the baseline. Moreover, the authors define the auxiliary task with a tradeoff parameter generally set to 1. It is not clear whether this is an appropriate choice and the authors have not provided the loss over training for each task. It is possible the auxiliary task loss is dwarfed by the central loss. Supp Figure 1C shows a dramatic influence of the tradeoff parameter on beta-lac performance. Would the authors be willing to show losses across each task and the corresponding aux task in the supplement?

(6) Size effects influence model predictivity. Supp Fig 1A indicates on different architectures that longer truncation lengths enable improved predictive performance. How do the authors suggest comparing performance given variable length inputs?

**Additional Feedback:**

1) When performing computations for PPI affinity or PLI, how do the authors account for charge states and other nuances sensitive to the environment that the protein is in? Protonation states and subtle changes to the molecular representation can have vastly different implications for whether there will be an interaction. Notes on how this is handled is important in all the benchmark tasks, as it will standardize how users would represent the input data (unless you are offering pre-processing in your code base?).

2) How did the authors populate feature vectors for the GIN? I could not find explicit details on how the ligands were input to the GIN for a representation.

3) It is curious that the DDE-based predictor did so well on certain tasks (namely the function tasks in Table 2). Would the authors be willing to address their hypothesis as to why? This representation seems to do reasonably well on almost all tasks except localization.

4) Table 3 indicates an unusual trend; the inclusion of a structure auxiliary task seems to diminish performance on a structure central task in more cases than it helps. Do the authors have an intuition why this is the case? This is actually a very interesting result, as one should expect these tasks to be beneficial to one another.

5) The MTL analysis does open a very interesting line of inquiry around what specific classes/improvements the auxiliary task confers in prediction. It would be nice to see (perhaps in follow up work) what classes do better/see reduced uncertainty in inference given the inclusion of the structural information. For some models, attention analysis etc. may open a new line of inquiry.

**Clarity:**

There is a minor typo on 5.3 under the bullet point "MTL less benefits..."; this might be "least benefits".

Impact of each of these datasets should be stated clearly. The authors explain what the datasets are, and what the underlying task is, but do not illustrate their rationale for why they included this task over other comparable datasets. Other such dataset or benchmarking papers often include an "impact" or "motivation" as to what attributes within the data make it attractive to include in a benchmark.

**Documentation:**

The supplemental information does not provide a comprehensive plan for growing and maintaining an open-source community.

(1) How do you plan to keep up to date with updates between common ML packages (ex torch/tensorflow compatibility)?

(2) Are there tutorials on how to run a custom model against this benchmarking dataset?

(3) The documentation seems to be a work-in-progress, but is very critical to accessibility. Moreover, the ability to train or test custom models on a subset of tasks will be an important (ex: some models, like the DDE feature model, cannot train on certain tasks).

(4) There is no contributions guide, or plan around how to invite collaborators, or interface with the broader research community.

Without these core elements, these benchmarks do not distinguish themselves from the repos/data stores that already contain the original source information of the tasks.

**Relation To Prior Work:**

As mentioned in weaknesses, there is no novelty introduced in the processing of the tasks. The benchmarks also do not include the SOTA models performances that currently exist for some of the provided tasks. This is an important point of comparison.

**Summary And Contributions:**

This paper compiles a list of 14 tasks across various protein datasets to serve as a potential benchmark in training sequence-based protein models. These tasks focus on a range of biological properties of interest. The authors aggregate these tasks into five branches: function, localization, structure, protein-protein interaction (PPI), and protein-ligand interaction (PLI).

The authors explore this collection of datasets by testing a variety of different model architectures on single-task and a novel multi-task framework. Their multi-task results include structure-based auxiliary tasks, under the assumption that "structure influences function" (which is a biologically appropriate assumption).

Finally, the authors provide a work-in-progress repo that should provide both pre-processing to the datasets, and a training script that allows users to verify their own model's performance against these datapoints.

---

> ### Author Response · Authors · 2022-08-09
> **Author Feedbacks to Reviewer 7t4f (Part 1/5)**
>
> >**Q1: Several of the function datasets are not generalizable. It is surprising the datasets proposed by FLIP are not included in these tasks.**
>
> Thanks for your comments. We agree that some of the function datasets are not generalizable and it is difficult to decide whether a set of function tasks is complete to test the capacity of models. In fact, this is exactly the reason why we need to include different function prediction tasks and different dataset splits with the aim that they can complement each other to test the generalization of models.
>
> Therefore, to make our benchmark more comprehensive, following your suggestions, we have included the three benchmark tasks of FLIP into our revised version and evaluated all baselines on them. The results of single-task learning have been added in Table 3. We are currently running experiments on multi-task learning and will update the results when finished.
>
> >**Q2: The authors should consider redefining balanced subsets for train/val/test. Alternatively, AUPRC/sensitivity or other metrics that explicitly consider imbalance should be reported.**
>
> **Table A: Sample distribution over classes on binary classification tasks.**
>
> |#Samples|Train: class 0|Train: class 1|Valid: class 0|Valid: class 1|Test: class 0|Test: class 1|
> |:----:|:----:|:----:|:----:|:----:|:----:|:----:|
> |Solubility prediction|36,403|26,075|4,045|2,897|999|1,000|
> |Binary localization prediction|2,107|3,077|703|1,026|743|1,006|
> |Yeast PPI prediction|2,522|2,423|39|56|185|209|
> |Human PPI prediction|17,414|18,255|168|147|119|118|
>
> **Table B: Residue distribution over classes on secondary structure prediction.**
> |#Residues|Train: class 0|Train: class 1|Train: class 2|Valid: class 0|Valid: class 1|Valid: class 2|Test: class 0|Test: class 1|Test: class 2|
> |:----:|:----:|:----:|:----:|:----:|:----:|:----:|:----:|:----:|:----:|
> |Secondary structure prediction|799,086|475,456|946,965|207,237|114,918|234,776|49,182|31,970|63,208|
>
> **Table C: Top 3 sample numbers over classes on multi-class classification tasks.**
>
> |#Samples|Train: Top 3 most|Train: Top 3 least|Valid: Top 3 most|Valid: Top 3 least|Test: Top 3 most|Test: Top 3 least|
> |:----:|:----:|:----:|:----:|:----:|:----:|:----:|
> |Subcellular localization prediction|[2426, 1635, 1185]|[93, 192, 214]|[809, 545, 395]|[31, 65, 72]|[808, 508, 393]|[30, 64, 70]|
> |Fold classification|[981, 401, 361]|[1, 1, 1]|[19, 18, 15]|[1, 1, 1]|[30, 28, 28]|[1, 1, 1]|
>
> &emsp;
>
> Thanks for pointing out this important aspect. In Tables A and B, we show that the number of samples over different classes can be regarded as **balanced** on **four binary classification tasks** and also on the **secondary structure prediction task**. Therefore, it is appropriate to use accuracy as the evaluation metric on these tasks.
>
> However, on **two multi-class classification tasks**, we do observe **class imbalance**, as shown in Table C. In particular, on fold classification, the three smallest classes of training, validation and test splits all contain only one sample, and thus the task is highly class-imbalanced. Based on this fact, we report the results on these two tasks in the Supplement-Section 4 with weighted F1, which is a metric designed for multi-class classification with label imbalance. We observe that the ranking of baselines under weighted F1 is almost unchanged compared to that under accuracy, where shallow CNN is still the best model among models trained from scratch, and ESM-1b remains the SOTA model on these two tasks. Therefore, the conclusions in Section 5.2 of the main paper still hold.
> Considering **the consistency of experimental conclusions** and **the comparability with previous benchmark results in the literature** where accuracy is commonly reported, we still employ **accuracy** as the metric for these two tasks in the **main paper** and provide **weighted F1** performance in the **supplement**.

---

> > ### Author Response · Authors · 2022-08-09
> > **Author Feedbacks to Reviewer 7t4f (Part 2/5)**
> >
> > >**Q3: PLI-type tasks may be just as sensitive to the ligand representation as the sequence representation. How would you account for the protein-sequence contribution in this type of task?**
> >
> > This is a good question. As you suggested, the quality of the ligand representation could equally affect the results of PLI prediction tasks as the protein sequence representation. In our PEER benchmark, we keep the contribution of the ligand representation consistent across different protein sequence representation methods by constantly using a 4-layer Graph Isomorphism Network (GIN) with 256 hidden dimensions to extract the ligand representation (a widely-used molecule encoder in previous practices [a,b]), which **fixes the model capacity on the ligand feature extraction side**. In this way, a good protein sequence encoder is expected to extract informative protein representations that **imply the protein structure and further reveal the pattern of protein-ligand binding, together with the ligand representation from a fixed-capacity ligand encoder.** We deem that such a scheme is sufficient to fairly compare the relative strength of different protein sequence representation methods.
> >
> > [a] Strategies for pre-training graph neural networks. Hu et al., ICLR, 2020.
> >
> > [b] Infograph: Unsupervised and semi-supervised graph-level representation learning via mutual information maximization. Sun et al., ICLR, 2020.
> >
> >
> > >**Q4: Many of the datasets already exist as benchmarks on their own. Why does this unique set of 14 tasks confer the appropriate way to explore the representational strength of protein modeling?**
> >
> > As you said, the protein function and structure prediction tasks in our benchmark are involved in TAPE, and the protein localization prediction tasks are involved in DeepLoc. However, we argue that the proposed PEER benchmark is of **unique value as a comprehensive benchmark incorporating the evaluation of protein function/localization/structure prediction, PPI prediction and PLI prediction**. **A general-purpose protein sequence representation model is expected to consistently perform well on all these tasks**. Such versatility of a model indicates its effectiveness on more types of real-world problems, and this property can only be evaluated on our benchmark but not on previous ones dedicated to one or two types of tasks. To manifest this emphasis of model evaluation, we will maintain an integrated leaderboard on our [project website](https://torchprotein.ai/benchmark) to identify the top models that broadly perform well on all five types of benchmark tasks.
> >
> > >**Q5: The MTL results are overstated. Authors are suggested to show losses across the center task and the corresponding auxiliary task.**
> >
> > This suggestion is constructive. To better show the influence of MTL, in the Table 4 of revision (corresponding to the original Table 3), we have added a relative improvement column for each model to indicate how each task that the model plays against is benefited from multi-task learning. We also add a relative improvement row at the bottom to indicate how each model is benefited from using a specific auxiliary task. Based on these results, we observe that, by using contact prediction, fold classification and secondary structure prediction as the auxiliary task, **the performance of ESM-1b is improved by 4.94%, 1.49% and 1.98% relative to single-task learning**, respectively. **The CNN model achieves a 5.09% relative improvement when augmented by the contact prediction auxiliary task**. These improvements are non-trivial. By comparison, the Transformer model trained from scratch is not benefited by MTL. We thus **tune down our claim in Section 5.3 to suggest only CNN and ESM-1b as the candidate baseline models for future MTL research**.
> >
> > Following your suggestion, we also plot the training loss curves and validation metric curves under single- and multi-task learning settings in Figure 2 (b)(c) in the revised supplementary. The auxiliary task loss is somehow dwarfed by the center task loss due to our multi-task training routine, but it helps the optimization and generalization of models on the center task. We analyze the phenomenon in Section 5.4 of the supplement.
> >
> >  >**Q6: How do the authors suggest comparing performance given variable length inputs?**
> >
> > Thanks for the reminder on this important comparison detail. For all baseline models except for ESM-1b, we use the full protein sequence as the input on all tasks, and, because of the intrinsic limit on input sequence length, ESM-1b truncates those sequences with more than 1022 residues by keeping the first 1022 residues. Therefore, we guarantee that **different models are fairly compared with each other under their maximum input capacity**. We have added this experimental setup to the Section 5.1 of the revised main paper.

---

> > > ### Author Response · Authors · 2022-08-09
> > > **Author Feedbacks to Reviewer 7t4f (Part 3/5)**
> > >
> > > >**Q7: Concerns on presentation clarity: unclear statement of impacts of considered datasets and a minor typo in Section 5.3.**
> > >
> > > Thanks for your great suggestions. In the revised main paper, we have fixed the typo in Section 5.3 and added a statement of the impact of each benchmark dataset in Section 3.
> > >
> > > >**Q8: There is no novelty introduced in the processing of the tasks. The benchmarks also do not include the SOTA models’ performance.**
> > >
> > > We respectfully disagree with the reviewer on the first point. We argue that **our benchmark does have contributions on the preprocessing of PPI and PLI datasets**. For yeast and human PPI prediction, the raw datasets [c,d] do not provide dataset splits, and cross validation is commonly performed. By comparison, we split these datasets into train/valid/test splits based on sequence identity cutoffs (details are in Section 3.4). For PPI affinity prediction, the raw SKEMPI dataset [e] does not provide dataset splits, while we split it based on the number of mutations and simulate a multi-round protein binder design scenario (details are in Section 3.4). For PLI prediction on PDBBind, based on the raw PDBBind-2019 dataset [f], we further remove the sample redundancy between training and test splits and hold out another validation split from training based on sequence clustering results (details are in Section 3.5).
> > >
> > > Thanks for the great advice of adding SOTA performance from literature. In Table 3 of the revised main paper, we add a column at the end of the table to list the SOTA model and its performance from literature for each applicable task. These results help to compare the baseline models in our benchmark with the current best ones in this field.
> > >
> > > [c] Using support vector machine combined with auto covariance to predict protein--protein interactions from protein sequences. Guo et al., Nucleic Acids Research, 2008.
> > >
> > > [d] Large-Scale prediction of human protein-protein interactions from amino acid sequence based on latent topic features. Pan et al., Journal of Proteome Research, 2010.
> > >
> > > [e] SKEMPI: a Structural Kinetic and Energetic database of Mutant Protein Interactions and its use in empirical models. Moal et al., Bioinformatics, 2012.
> > >
> > > [f] Forging the basis for developing protein-ligand interaction scoring functions. Liu et al., Accounts of Chemical Research, 2017.
> > >
> > > >**Q9: The supplemental information does not provide a comprehensive plan for growing and maintaining an open-source community.**
> > >
> > > Appreciate for your great view on the open-source project.  We would like to share with you the latest progress of our PEER benchmark project, where the [project website](https://torchprotein.ai/benchmark) is mostly done. The source codes will be integrated into the [TorchDrug library](https://github.com/DeepGraphLearning/torchdrug/) and officially released together with the website at the end of this month.
> > >
> > > 1. **ML package compatibility**: Since the source codes of PEER benchmark will be available as a part of TorchDrug, these codes will be jointly maintained along with the TorchDrug library to make them compatible with several latest PyTorch versions.
> > > 2.  **Tutorial**: On the website, we have provided a [tutorial](https://torchprotein.ai/tutorial_2) on how to define a simple protein sequence encoder and solve various benchmark tasks with it. Users can also customize their own model by implementing a PyTorch-style model class and registering it in TorchDrug. [This TorchDrug tutorial](https://torchdrug.ai/docs/notes/model.html) gives an example of model customization, and we will write a similar tutorial for the PEER benchmark. After customizing the model, users can simply follow the guidance in the [first tutorial](https://torchprotein.ai/tutorial_2) to play against our benchmark tasks.
> > > 3. **Documentation**: After the release of source codes, the documentation of PEER benchmark will be merged into the [documentation of TorchDrug](https://torchdrug.ai/docs/). This documentation will cover the definitions of all the datasets, models and tasks used in our benchmark, which makes it easy for users to reproduce the benchmark results and customize their own models against benchmark tasks.
> > > 4. **Contribution guidelines**: The PEER benchmark will follow the [contribution guidelines of TorchDrug](https://torchdrug.ai/contribute) to gather the efforts from the community. In short, we will welcome all kinds of contributions, including submitting new benchmark results, implementing new datasets and models, improving documentation and fixing bugs.
> > > 5. **Code of Conduct**: The PEER benchmark will follow the [Code of Conduct of TorchDrug](https://github.com/DeepGraphLearning/torchdrug/blob/master/CODE_OF_CONDUCT.md) to act and interact in ways that contribute to an open, welcoming, diverse, inclusive, and healthy community.

---

> > > > ### Author Response · Authors · 2022-08-09
> > > > **Author Feedbacks to Reviewer 7t4f (Part 4/5)**
> > > >
> > > > >**Q10: When performing computations for PPI or PLI affinity, how do the authors account for charge states and other nuances sensitive to the environment that the protein is in?**
> > > >
> > > > Thanks sincerely for the helpful suggestion. Indeed, we were aware of the effects from environmental factors (the inputs except binding subjects: proteins or small molecules) can contribute a lot to the realistic binding equilibrium: solvation, pH, protonation state and even multiple binding pockets/poses, to name a few. For traditional structure-based complex modeling (e.g., docking), they should carefully attend to these factors since the structure-based models are trying to model the interactions under these environmental configurations. However, in our cases (PEER or other sequence-based benchmark works), this ever-important consideration can be not applicable, which will be explained below.
> > > > 1. First, we would like to overview the data source: all of 5 (3 for PPI, 2 for PLI) datasets are the collection as open data-source and from experimentally measured binding affinity data. For example, as stated in their [official website](http://www.pdbbind.org.cn/), PDBbind database is to provide a comprehensive collection of experimentally measured binding affinity data for all biomolecular complexes deposited in the Protein Data Bank (PDB). The authors provide structural data (in PDB or SDF format) of each binding partner, and they are responsible for the preprocessing procedure for cleaned complex records. In our benchmark, each PPI/PLI database is measured or maintained by the same group/groups, and they have made great efforts to ensure that the experimental environments and conditions are identical or as similar as possible among different data items. In this sense, readers can interpret that each PPI/PLI task aims to **make predictions under specific and definite experimental conditions that belong to that task**.
> > > > 2. Furthermore, we must admit that our input space representations in PEER (sequence for protein, 2D graph for small molecule) is in an approximate or noisy manner. To be specific, even presenting in the original dataset, much finer representations and the experimental conditions (such as atom-level 3D coordinates, pH, solvation-related parameters) are neither included as a part of model input in PEER nor distinguished under our input format. However, we would like to point out that this is **a common practice in recent deep learning-based works for drug discovery**. Since sequence representation itself is already an information-lost format for modeling/prediction, our main purpose in PEER is not to include/recover as many input features as possible (even they are important and necessary); instead, it aims to **depict a specific type of energy landscape (in binding case, the Kd/Ki/IC50 values) existing in the data space**, where **the use of insufficient input parameters can be tolerated**. As an analogy, **AlphaFold never accounts for the solvation effect when folding a protein**; **any docking program never accounts for the states of valence electrons on the molecular orbitals**.
> > > >
> > > > >**Q11: How did the authors populate feature vectors for the GIN?**
> > > >
> > > > Thanks for the reminder on this implementation detail. Following the featurization scheme of TorchDrug, we represent ligand inputs by a set of atom-level features, referring to [this code implementation](https://github.com/DeepGraphLearning/torchdrug/blob/dbf6f4e903308807608818cf77b1ac2d879815cb/torchdrug/data/feature.py#L50). Specifically, for each atom of an input ligand, we construct its input features by concatenating the one-hot embeddings of atomic symbol, atomic chirality tag, atom degree, formal charge, number of hydrogens on the atom, number of radical electrons on the atom, the atom's hybridization, binary aromatic flag and binary within-a-ring flag. These one-hot embeddings form 66-dimensional atom features as the input node features of GIN. We have supplemented these implementation details to the “Model setups” part of Section 5.1 in the revision.

---

> > > > > ### Author Response · Authors · 2022-08-09
> > > > > **Author Feedbacks to Reviewer 7t4f (Part 5/5)**
> > > > >
> > > > > >**Q12: Why could the simple DDE-based predictor perform well on almost all tasks?**
> > > > >
> > > > > The DDE feature descriptor is essentially constructed based on the frequency of different dipeptides in the protein sequence. In the literature of computational biology, the features based on dipeptide frequency have demonstrated their effectiveness on a variety of protein understanding tasks, including protein stability analysis [g], protein solubility prediction [h] and protein structure prediction [i], etc. Based on these broad successes of applying dipeptide frequency features to protein understanding tasks, the effectiveness of DDE-based predictor on our benchmark tasks is also foreseeable.
> > > > >
> > > > > [g] Correlation between stability of a protein and its dipeptide composition: a novel approach for predicting in vivo stability of a protein from its primary sequence. Guruprasad et al., Protein Engineering, Design and Selection, 1990.
> > > > >
> > > > > [h] PROSO II--a new method for protein solubility prediction. Smialowski et al., The FEBS journal, 2012.
> > > > >
> > > > > [i] Prediction of protein structural classes using support vector machines. Sun et al., Amino Acids, 2006.
> > > > >
> > > > > >**Q13: The inclusion of a structure auxiliary task seems to diminish performance on a structure central task in more cases than it helps. Why is this the case?**
> > > > >
> > > > > This is a good question. We explain the experimental results for the MTL within the structure prediction group in two parts as below.
> > > > >
> > > > > 1. **Why fold classification benefits from contact and secondary structure prediction, but the reverse is not true?** Note that fold classification is a protein-wise classification task and requires graph representations to capture structural patterns, while contact and secondary structure prediction are residue-level tasks and require residue representations to reflect (relative) positional and structural information. Therefore, the latter two tasks need much finer representations than the former. It could be beneficial to include supervision on the finer level when learning protein representations, but it may be harmful to put effort into coarser-level representations when learning finer-level representations.
> > > > > 2. **Why does contact prediction help secondary structure prediction, but the reverse is not true?** This can be understood that two residues in contact are likely to belong to the same secondary structure, but not all residues in the same secondary structure are in contact, e.g., two end residues of a very long alpha helix.
> > > > >
> > > > > >**Q14: It would be nice to see (perhaps in follow-up works) what classes do better/see reduced uncertainty in inference given the inclusion of the auxiliary structural information.**
> > > > >
> > > > > Appreciate for this interesting research direction. Mining the improvements of model prediction brought by auxiliary structural information definitely deserves further studies. We will place it as one of our major future directions.

---

> ### Author Response · Authors · 2022-08-22
> **Follow-up Feedbacks to Reviewer 7t4f**
>
> Dear Reviewer 7t4f,
>
> Thanks again for your great review.
>
> We have now **finished all the single- and multi-task learning experiments on three function prediction tasks from FLIP**, and these benchmark results have been added to the **Table 3 and 4 of the revised main paper**. Particularly, for multi-task learning experiments, the performance of CNN is enhanced on all three FLIP tasks by using all three considered auxiliary tasks, and the performance of Transformer on FLIP is also improved under most multi-task settings. Please check them out.
>
> We welcome any further questions. The newest progress of our project will be updated to you on OpenReview.
>
> &emsp;
>
> Best regards,
>
> Authors of Paper #64

---

> > ### Comment · Reviewer_7t4f · 2022-08-24
> > **Thank you for your response**
> >
> > Dear authors,
> >
> > I am hesitant to change my review for the following reasons:
> >
> > - Lack of novelty: It seems 15/17 tasks maintain the pre-processing of the original authors who published the dataset. While it's useful to have a consolidated location to access these datasets, it relies on the assumptions of the original authors who may not have been thinking of its use in a broader context. Thus this may mean that that certain datasets are incompatible, irrelevant, or worse yet, detrimental. The MTL results seem to suggest slight gains at best on the subset of tasks performed. Thus, it is hard to say if these datasets probe a set of consistent rules (i.e. "grammar" or biophysical properties), given an architecture, that make for a good protein sequence representation (one might envision stronger MTL results if there was). It is worth discussing with the original authors of the datasets what the strengths and weaknesses are, and explicitly highlighting this. Moreover, the tasks do not cover more recent interests for example, prediction of 3D structure, disordered regions, kinetics, etc.
> >
> > - "Good performance" does not necessarily translate to biological principles: There is no clear direction on how a biologist may interpret highly performing models on a benchmark, because the tasks implicitly characterize a system/temperature/pH/charge. A small boost in performance on the benchmarks could be statistically significant, but be entirely irrelevant in reflecting appropriate biological properties. Benchmarking should ensure good performance is not a result of something trivial (like massive numbers of parameters).
> >
> > - Biological relevance: If the goal of the paper is to inquire why a protein sequence representation is good, this requires the ability to ask more meaningful questions of the datasets beyond performance. Namely, it should reveal how inductive biases may assist in making more robust predictions. For example, when a benchmark improves -is it classifying more of the rare event classes? Or just getting a few more examples of the majority class? What about these tasks give me confidence that a model architecture could generalize to a new domain? What is the uncertainty in the prediction?
> >
> > I think this work is an important step in the right direction; I commend the authors for pursuing this avenue of research.

---

### Author Response · Authors · 2022-08-09
**Author Responses to All Reviewers**

Dear Reviewers,

We first would like to appreciate the constructive suggestions and golden comments from you. These suggestions and comments will definitely enhance the quality of our paper and project.

We have posted the first version of responses and paper revision. We hope these responses can answer your questions, and we welcome any further questions. In the revision of main paper and supplementary material, we use **RED texts** to denote **the revisions made based on your great suggestions**. Please check them out!

&emsp;


Best regards,

Authors of Paper #64

---

### Comment · Reviewer_T5Wq · 2022-08-28
**need a bit more novelty in the  benchmarking design**

In this manuscript, the authors present PEER, a collection of protein ML-related tasks with pre-defined splits, covering five critical categories to test ML models’ understanding of proteins. The authors also benchmarked multiple protein language models (LMs) and reported performances on the PEER tasks.

Overall, I recommend PEER add more novelty to benchmarking design.

The current version is an amalgamation of a few popular protein ML benchmarks, including TAPE, FLIP, and PDBBind. I recommend the authors add more original contributions to improving the practical impact of those tasks.

A few more specific comments:

1. Related work “Protein modeling benchmarks” is missing some important work:

- Therapeutics Data Commons: Machine Learning Datasets and Tasks for Drug Discovery and Development

- Add TDA https://arxiv.org/abs/2102.09548: contains several related protein tasks such as PPI and protein-ligand interactions.

- Atom3d

2. Comments regarding curated tasks:

- Metrics such as accuracy and RMSE for PPI and PLI are inappropriate. In practical use cases such as protein design and virtual screening, ranking metrics such as precision@k are more appropriate.

- It is not stated in the manuscript how the negative pairs are produced in some of the PPI and PLI datasets. This is important since our current knowledge about positive PPI/PLI pairs are partial observation of the collection of ground truths.

Resources like http://mips.helmholtz-muenchen.de/proj/ppi/negatome/ would be helpful.

- Formuating “Fold classification” task as a 1,195-class categorical classification is probably not appropriate as the SCOP classification of protein fold has hierarchical structure.

- Homology-related tasks are missing from the collection. This is important to validate the promise that protein LMs can learn evolutionary information.

---

> ### Author Response · Authors · 2022-08-29
> **Author Feedbacks to Reviewer T5Wq**
>
> >**Q1: Some important related works are missed.**
>
> Thanks for pointing out these important related works. In the revised main paper, we have discussed TDA [a] and ATOM3D [b] as protein-related benchmarks. In the future, we will definitely consider adding the PPI and PLI tasks in TDA to our PEER benchmark.
>
> >**Q2: Metrics such as accuracy and RMSE for PPI and PLI are inappropriate.**
>
> We agree that **ranking metrics** are better choices in a practical protein binder design application where **the target protein is fixed and the binder is mutated**. However, in **our benchmark datasets** (SKEMPI, PDBbind and BindingDB), there are **hundreds/thousands of target proteins**, and the model is asked to predict the binding affinity of the combinations with all these proteins. In such a scenario, regression metrics (*e.g.*, RMSE) are more appropriate and commonly-used [c,d] than ranking metrics. However, considering the practical importance of selecting top binders for a specific protein, we will supplement such tasks and evaluate the ranking performance in the following works of this project.
>
> >**Q3: It is not stated in the manuscript how the negative pairs are produced in some of the PPI and PLI datasets.**
>
> For Yeast and Human PPI prediction tasks, we select negative pairs by randomly selecting proteins from different subcellular locations. We have added the statement of this scheme to the revision.
>
> Thanks so much for pointing out the [Negatome Database 2.0](http://mips.helmholtz-muenchen.de/proj/ppi/negatome/). The negative pairs in this database are with higher quality and less biased compared to the random sampling from different subcellular locations. We will try to combine our currently used negative pairs with the ones from this resource to further debias two PPI benchmark datasets.
>
>  >**Q4: Formulating “Fold classification” task as a 1,195-class categorical classification is probably not appropriate.**
>
> Indeed, the SCOP classification system classifies proteins hierarchically on 7 different levels. However, **the classification task performed on the “fold level” is of most research interests**, which seeks to identify protein homologs sharing the same fold [e] and can further benefit protein structure [f] and function [g] prediction. Therefore, in our benchmark, we only include the fold-level classification task for its broad research interests.
>
> >**Q5: Homology-related tasks are missing from the collection.**
>
> We would like to argue that **the fold classification task is actually a homology-related task**. This task intends to assign SCOP fold-level labels to protein sequences. For dataset splitting, we hold out entire superfamilies from training to compose the test set, allowing us to evaluate the ability of the model on detecting proteins with similar structures but dissimilar sequences, *i.e.*, performing remote homology detection. Of course, there exist other important homology-related tasks, *e.g.*, the retrieval of homologous proteins by directly comparing the protein embedding similarity. We will explore these tasks in the future works of our benchmark project.
>
> &emsp;
>
> [a] Therapeutics data commons: Machine learning datasets and tasks for drug discovery and development. Huang, Kexin, et al., arXiv preprint arXiv:2102.09548 (2021).
>
> [b] Atom3d: Tasks on molecules in three dimensions. Townshend, Raphael JL, et al., arXiv preprint arXiv:2012.04035 (2020).
>
> [c] K deep: protein–ligand absolute binding affinity prediction via 3d-convolutional neural networks. Jiménez, José, et al., Journal of chemical information and modeling 58.2 (2018): 287-296.
>
> [d] Structure-aware interactive graph neural networks for the prediction of protein-ligand binding affinity. Li, Shuangli, et al., Proceedings of the 27th ACM SIGKDD Conference on Knowledge Discovery & Data Mining. 2021.
>
> [e] DeepSF: deep convolutional neural network for mapping protein sequences to folds. Hou, Jie et al., Bioinformatics 34.8 (2018): 1295-1303.
>
> [f] Improving protein fold recognition by deep learning networks. Jo, Taeho, et al., Scientific reports 5.1 (2015): 1-11.
>
> [g] Integrated protein function prediction by mining function associations, sequences, and protein–protein and gene–gene interaction networks. Cao, Renzhi et al., Methods 93 (2016): 84-91.

---

### Meta-Review · Area_Chair_wLfB · 2022-09-14

**Recommendation:** Accept
**Confidence:** 4

**Metareview:**

This work presents a diverse benchmark for multiple proteomic tasks, including comprehensive results for both single-task and multi-task learning. The paper is well written, and the benchmark is essential to the field. The mean reviewer's score is near acceptance. Perhaps most importantly, the reviewer's comments have been addressed during the rebuttal, and the code has been made publicly available. Therefore, I decide to accept this paper.

An additional helpful baseline would include a generic implementation of a system of neural networks that performs multiple tasks: consisting of (i) encoder networks for each input; (ii) a bottle-neck fusion network with fully connected layers that learns how to merge the encoded representations; and (iii) decoder networks for each output.

---

### Decision · Program_Chairs · 2022-09-16

Accept